# On the Entropy Dynamics in Reinforcement Fine-Tuning of Large Language Models

**Shumin Wang** [1 †]  **Yuexiang Xie** [2]  **Wenhao Zhang** [2]  **Yuchang Sun** [2]  **Yanxi Chen** [2]  **Yaliang Li** [2]  **Yanyong Zhang** [1]

## Abstract

Entropy serves as a critical metric for measuring the diversity of outputs generated by large language models (LLMs), providing valuable insights into their exploration capabilities. While recent studies increasingly focus on monitoring and adjusting entropy to better balance exploration and exploitation in reinforcement fine-tuning (RFT), a principled understanding of entropy dynamics during this process is yet to be thoroughly investigated. In this paper, we establish a theoretical framework for analyzing the entropy dynamics during the RFT process, which begins with a discriminant expression that quantifies entropy change under a single logit update. This foundation enables the derivation of a first-order expression for entropy change, which can be further extended to the update formula of Group Relative Policy Optimization (GRPO). The corollaries and insights drawn from the theoretical analysis inspire the design of entropy control methods, and also offer a unified lens for interpreting various entropy-based methods in existing studies. We provide empirical evidence to support the main conclusions of our analysis and demonstrate the effectiveness of the derived entropy-discriminator clipping methods. This study yields novel insights into RFT training dynamics, providing theoretical support and practical strategies for optimizing the exploration-exploitation balance during LLM fine-tuning. We release our code at https://github.com/agentscope-ai/Trinity-RFT/tree/main/examples/entropy.

†Work done as an intern at Tongyi Lab, Alibaba Group. [1]University of Science and Technology of China [2]Tongyi Lab, Alibaba Group. Correspondence to: Yanyong Zhang <yanyongz@ustc.edu.cn>, Yaliang Li <yaliang.li@alibaba-inc.com>.

*Proceedings of the 43rd International Conference on Machine Learning*, Seoul, South Korea. PMLR 306, 2026. Copyright 2026 by the author(s).

## 1. Introduction

Reinforcement fine-tuning (RFT) (OpenAI, 2025) has recently attracted growing attention as a post-training paradigm for enhancing the capabilities of large language models (LLMs) (Guo et al., 2025; Yang et al., 2025a; Agarwal et al., 2025). It has shown substantial improvements across a range of downstream tasks, such as mathematical reasoning (Shao et al., 2024; Chen et al., 2025a), programming (Wei et al., 2025; Zeng et al., 2025), and tool usage (Zhang et al., 2025; Feng et al., 2025).

Drawing from reinforcement learning (RL) (Sutton et al., 1998), RFT transforms the fine-tuning process into a policy optimization problem where LLMs are incentivized to produce high-reward responses. The exploration-exploitation trade-off presents a crucial challenge for RFT, potentially leading to unstable performance and stagnation in local optima (Arulkumaran et al., 2017; Ahmed et al., 2019). In this context, the *entropy* of responses emerges as a key diagnostic metric, offering insights into the output diversity of LLMs, and is actively leveraged by recent studies (Yu et al., 2025; Cui et al., 2025; Hu et al., 2025; Su et al., 2025) to monitor training dynamics and regulate policy behavior.

However, existing methods (Wang et al., 2025; Liao et al., 2025; Yu et al., 2025; He et al., 2025) often rely on heuristic designs that treat entropy in isolation and oversimplify its adjustment. Moreover, the divergence in whether these approaches encourage or suppress entropy highlights a fundamental lack of in-depth understanding of entropy dynamics (Hu et al., 2025; An et al., 2025). Such an unprincipled basis can lead to labor-intensive hyperparameter tuning without clear guidance, thus hindering the effective optimization of RFT. As a result, a theoretically grounded framework is increasingly necessary to characterize entropy dynamics in RFT.

To fill this gap, we establish a theoretical framework that provides a principled understanding of entropy dynamics in RFT. Inspired by (Ren & Sutherland, 2025), we model the update of a single token's logit during optimization, and characterize how it propagates through the model's output probability distribution, ultimately influencing the policy's entropy. Our derivation reveals that the entropy

change direction is determined by the interplay between the update direction (whether the token is rewarded or penalized) and the sign of the proposed discriminator score $S_*$, which captures the relationship between token probability and policy entropy. This analysis explains the widely observed phenomenon of rapid entropy collapse (Yu et al., 2025) when models are consistently rewarded for generating high-probability and "safe" responses (Su et al., 2025).

Building upon such single-logit analysis, we extend our framework to analyze the entropy change resulting from an optimization step under Group Relative Policy Optimization (GRPO) (Shao et al., 2024). We derive an expression that practically computes the entropy change trend leveraging the discriminant and its policy-weighted expectation. Our analysis provides insights for the development of entropy-based methods, inspires practical clipping strategies and sheds light on the mechanisms of existing approaches.

Our contributions can be summarized as follows:

- We propose a theoretical framework that characterizes the token-level entropy change during policy optimization. We further extend it to a practical GRPO optimization step and derive a first-order analytical expression, indicating that the direction of entropy change is closely related to the direction of token updates and a discriminator score $S_*$.

- Our theoretical analysis provides new insights for the design of entropy control methods. Building upon this, we explain existing entropy control methods from the perspective of entropy dynamics, offering a unified and principled theoretical framework for understanding their effects and underlying mechanics.

- We conduct experiments to provide empirical evidence for our theoretical analysis, showing that $S_*$ can be a reliable discriminator for the entropy dynamics. The experimental results also demonstrate the effectiveness of the derived clipping methods in stabilizing the entropy in RFT to promote model exploration.

## 2. Preliminaries

**Group Relative Policy Optimization (GRPO)** GRPO (Shao et al., 2024) is a prominent RFT algorithm that has proven highly effective and efficient across various tasks (Guo et al., 2025; Zhang et al., 2025). In GRPO, for each query $q$, a behavior policy $\pi_{\theta_{sample}}$ is employed to sample a group of $G$ responses $\{o_i\}_{i=1}^{G}$, where each response $o_i = (a_{i,1}, \ldots, a_{i,T_i})$ with $T_i$ tokens is subsequently assigned a scalar reward $R_i$, and each token $a_{i,t}$ in response is generated under state $s_{i,t} = (q, o_{i,<t})$. The policy is updated by maximizing the GRPO objective

function, defined as:

$$\mathcal{J}_{\text{group}}(\theta) = \frac{1}{\sum_{j=1}^{G} T_j} \sum_{i=1}^{G} \sum_{t=1}^{T_i}$$
$$\min\left(r_{i,t}(\theta)A_i, \, \text{clip}\left(r_{i,t}(\theta), 1 - \varepsilon_l, 1 + \varepsilon_h\right)A_i\right). \quad (1)$$

Following (Yu et al., 2025; An et al., 2025; Wang et al., 2025), we omit the KL divergence penalty. Here, the advantage $A_i$ is computed by standardizing rewards within the group, and the importance ratio $r_{i,t}(\theta)$ is the token-level probability ratio between the target and behavior policies, i.e., $A_i = \frac{R_i - \text{mean}(\{R_j\}_{j=1}^{G})}{\text{std}(\{R_j\}_{j=1}^{G})}$ and $r_{i,t}(\theta) = \frac{\pi_\theta(a_{i,t}|s_{i,t})}{\pi_{\theta_{sample}}(a_{i,t}|s_{i,t})}$. The parameter $\varepsilon$ defines the clipping range for PPO-style (Schulman et al., 2017) clipped objective. In our "strict on-policy training" (Chen et al., 2025a) setup, where the behavior policy is the optimized policy ($\pi_{\theta_{sample}} = \pi_\theta$), the importance ratio satisfies $r_{i,t} = 1$ and the clipping mechanism remains inactive. In this case, the update of GRPO encourages increasing the probability of sampled tokens if $A_i > 0$ and decreasing it if $A_i < 0$.

**Entropy Dynamics** Entropy provides a principled measure of uncertainty and is used to quantify the diversity of model outputs. For an LLM, the next-token distribution is given by $\mathbf{p}_t(\cdot) = \pi_\theta(\cdot \mid s_t) = \text{softmax}(\mathbf{z}_t)$, where $\mathbf{z}_t$ are the model's logits at position $t$ in a response. The token-level entropy is then defined as $H_t = -\sum_{i \in [V]} p_i^t \log p_i^t$, where $V$ denotes the size of vocabulary $\mathcal{V}$ and $p_i^t$ is the probability of token $a_t$ being the $i$-th vocabulary item.

The field of *learning dynamics* (Ren & Sutherland, 2025) studies how parameter updates affect model predictions. In this work, we introduce the concept of *Entropy Dynamics*, focusing specifically on how token entropy evolves during RFT. Specifically, we investigate how a parameter update, triggered by a single sampled token $a_t$, alters the entropy of the output token distribution at that step.

We formalize this by investigating the relationship between the entropy change before and after update, $\Delta H_t$, and the distribution of the policy in position t, $\pi_\theta(a_t \mid s_t)$. By analyzing this relationship, we aim to uncover the principles that determine whether the updates in RFT encourage diverse responses or lead to repetitive, similar outputs.

## 3. Analysis of the Entropy Dynamics in RFT

To establish a principled understanding of entropy dynamics in RFT, we propose a theoretical framework that characterizes the token-level entropy change during policy optimization. Specifically, we quantify how the update of a single token affects the policy's entropy, providing a microscopic view of entropy dynamics. Upon this, we derive the first-order expression for the entropy change resulting from a

policy update step when applying GRPO.

### 3.1. From a Single Logit Update to the Entropy Change

We consider a single decoding step where the policy $\pi$ produces a distribution over a vocabulary $\mathcal{V}$ of size $V$. Let $\mathbf{z} \in \mathbb{R}^V$ be the model's output logits. These logits are transformed into a probability distribution $\mathbf{p}$ via the softmax function, where $p_i = \frac{\exp(z_i)}{\sum_{j=1}^{V} \exp(z_j)}, \forall i \in [V]$. The diversity of this probability distribution is measured by the token-level Shannon entropy (Shannon, 1948), which can be formally given as $H(\mathbf{p}) = -\sum_{i=1}^{V} p_i \log p_i$.

Throughout our analysis, we make the following standard assumptions for deriving first-order dynamics: (i) All probabilities $\{p_i\}$ are non-zero, as guaranteed by the softmax function; (ii) Auxiliary regularization terms, including KL-divergence penalties, and explicit entropy bonuses, are considered inactive within RFT unless explicitly specified; and (iii) We ignore tokens that trigger logit clipping as their gradients are set to zero and contribute no change to entropy.

The analysis begins with a fundamental operation in model update, i.e., updating the logit of a single token. We model this as a perturbation as $\delta \mathbf{z} = \varepsilon \cdot \mathbf{e}_k$, where $\mathbf{e}_k$ is the standard basis vector for the $k$-th token, and $\varepsilon$ is the change caused by the optimization process. The sign of $\varepsilon$, i.e., $\mathrm{sign}(\varepsilon)$, represents the direction of the update: $\mathrm{sign}(\varepsilon) = +1$ corresponds to rewarding the token (increasing the logits), while $\mathrm{sign}(\varepsilon) = -1$ corresponds to penalizing it (decreasing the logits). The following lemma quantifies how this logit perturbation propagates to the probability distribution.

**Lemma 3.1.** *Given a logit perturbation $\delta \mathbf{z} = \varepsilon \cdot \mathbf{e}_k$ on $k$-th token $a^k$ in the vocabulary, the resulting first-order change in the probability distribution $\mathbf{p}$ is given by:*

$$\delta p_k = \varepsilon p_k(1-p_k) \text{ and } \delta p_i = -\varepsilon p_i p_k, \forall i \in [V], i \neq k. \quad (2)$$

*Proof.* The Jacobian of the softmax function is $\frac{\partial p_i}{\partial z_j} = p_i(\mathbf{1}\{i = j\} - p_j)$, where $\mathbf{1}\{\cdot\}$ is the indicator function. The first-order change $\delta p_i$ is given by the Taylor expansion $\delta p_i = \sum_{j=1}^{V} \frac{\partial p_i}{\partial z_j} \delta z_j + O(\varepsilon^2)$. Since $\delta z_j = \varepsilon \cdot \mathbf{1}\{j = k\}, \forall j \in [V]$, we have $\delta p_i = \frac{\partial p_i}{\partial z_k} \varepsilon = \varepsilon \cdot p_i(\mathbf{1}\{i = k\} - p_k)$, which yields the results in (2). $\square$

An immediate consequence of Lemma 3.1 is that the relative change in probability is uniform for all unperturbed tokens. Based on (2), the changes in probabilities are given by:

$$\frac{\delta p_k}{p_k} = \varepsilon(1-p_k) \text{ and } \frac{\delta p_i}{p_i} = -\varepsilon p_k, \forall i \in [V], i \neq k. \quad (3)$$

The analysis shows that, when the probability of token $a^k$ is adjusted, its probability mass is redistributed proportionally

from (or to) all other tokens. This aligns with the observation in previous works (Ren & Sutherland, 2025).

Building upon this insight, we can now derive a closed-form expression for the first-order change in entropy. We first define a key quantity that would determine the direction of this change. Let the entropy change discriminator for token on position $t$ be defined as $S_i^t \triangleq p_i^t(H^t + \log p_i^t)$, where the subscript $t$ is omitted when not causing confusion. In particular, assuming token $a^k$ is chosen at this position, the corresponding discriminator is denoted as $S_*^t \triangleq S_k^t$.

**Theorem 3.2.** *The first-order change in entropy, denoted by $\Delta H$, under the perturbation $\delta \mathbf{z} = \varepsilon \mathbf{e}_k$ is given by:*

$$\Delta H = -\varepsilon S_* + O(\varepsilon^2). \quad (4)$$

*Proof.* The first-order Taylor expansion of entropy $H$ around $\mathbf{p}$ can be given by:

$$\Delta H = H(\mathbf{p}+\delta\mathbf{p}) - H(\mathbf{p}) = \sum_{i=1}^{V} \frac{\partial H}{\partial p_i} \delta p_i + O(\|\delta\mathbf{p}\|^2). \quad (5)$$

Since $\frac{\partial H}{\partial p_i} = -(1 + \log p_i)$ and conservation of probability implies $\sum_i \delta p_i = 0$, we have:

$$\Delta H = -\sum_{i=1}^{V} (1 + \log p_i)\delta p_i + O(\varepsilon^2) \quad (6)$$

$$= -\sum_{i=1}^{V} \log p_i \delta p_i + O(\varepsilon^2). \quad (7)$$

Substituting the expressions for $\delta p_i$ from Lemma 3.1, $\Delta H$ can be simplified as:

$$\Delta H = -\varepsilon p_k\big((1 - p_k)\log p_k - \sum_{i \neq k} p_i \log p_i\big) + O(\varepsilon^2)$$
$$= -\varepsilon p_k\big(H + \log p_k\big) + O(\varepsilon^2),$$

which completes the proof. $\square$

**Implications** Theorem 3.2 provides a simple yet effective criterion for determining how a single-token update affects policy entropy. The direction of entropy change is dictated by the sign of two factors: the update direction $\varepsilon$ and the discriminator $S_*$. The sign of the discriminator $S_*$ depends on the relationship between the token's probability $p_k$ and the overall entropy $H(\mathbf{p})$:

$$\mathrm{sign}(S_*) = \mathrm{sign}\left(H(\mathbf{p}) + \log p_k\right) = \mathrm{sign}(p_k - e^{-H(\mathbf{p})}).$$

Consequently, rewarding a token $(\mathrm{sign}(\varepsilon) = +1)$ increases entropy if its probability $p_k < e^{-H(\mathbf{p})}$ (a relatively low-probability token) and decreases entropy if $p_k > e^{-H(\mathbf{p})}$ (a relatively high-probability token). The relationship is reversed when a token is penalized $(\mathrm{sign}(\varepsilon) = -1)$.

This microscopic analysis is the foundational building block for understanding entropy dynamics in RFT. Given that

most existing RFT algorithms (Shao et al., 2024; Yu et al., 2025; Zheng et al., 2025) apply an update signal of the same direction to all tokens within a single response, our analysis explains the common empirical observation of rapid entropy collapse when models are consistently rewarded for generating high-probability and "safe" responses (He et al., 2025), which can lead to a gradual loss of the model's exploratory capabilities.

## 3.2. Extension to a GRPO Optimization Step

Beyond the above single-logit analysis, we extend our framework to model the entropy change resulting from a GRPO optimization step introduced in Section 2. Recall the training objective function of GRPO in equation 1, for a chosen token $a^k$ with token id $k$, its contribution to the whole training target can be given as: $\frac{\mathbf{p}_k}{p'_k} \cdot A$, where $\mathbf{p}_k$ denotes the current model distribution in the sampled position, $p'_k$ is the probability of the sampled token under the sampling model distribution and $A$ represents its advantage. Therefore, its contribution to the training loss is given by a surrogate loss:

$$\mathcal{L}(\mathbf{z}) = r \cdot A \cdot \log p_k(\mathbf{z}), \tag{8}$$

where $r = \frac{\pi_\theta(a^k)}{\pi_{\theta_{\text{sample}}}(a^k)}$ is the importance sampling ratio.

A single gradient update step with learning rate $\eta$ results in a first-order change to the logits $\mathbf{z}$:

$$\delta\mathbf{z} = \eta \nabla_\mathbf{z} L = \alpha \nabla_\mathbf{z} \log p_k, \tag{9}$$

where we define $\alpha = \eta r A$ as the effective step size.

Recall the Jacobian of the softmax function in Lemma 3.1, $\nabla_\mathbf{z} p_k = p_k(\mathbf{e}_k - \mathbf{p})$. Therefore, we have:

$$\delta\mathbf{z} = \alpha \nabla_\mathbf{z} \log p_k = \alpha \frac{1}{p_k} \nabla_\mathbf{z} p_k = \alpha(\mathbf{e}_k - \mathbf{p}). \tag{10}$$

**Theorem 3.3.** *Let $S_i$ be the entropy discriminant for token $i$, and let its expectation over the policy distribution be $\mathbb{E}_{i\sim\mathbf{p}}[S_i] = \sum_{i=1}^V p_i S_i$. The first-order change in entropy of a token $H(\mathbf{p})$ satisfies:*

$$\Delta H = -\alpha (S_* - \mathbb{E}_{i\sim\mathbf{p}}[S_i]) + O(\alpha^2). \tag{11}$$

*Proof.* Recall the token-wise objective defined in (8) and a single update step defined in (9). Since $\mathbf{p} = \text{softmax}(\mathbf{z})$, its Jacobian matrix is $J = \frac{\partial \mathbf{p}}{\partial \mathbf{z}} = \text{diag}(\mathbf{p}) - \mathbf{p}\mathbf{p}^\top$, yielding the following equations:

$$\delta\mathbf{p} = J\delta\mathbf{z} = (\text{diag}(\mathbf{p}) - \mathbf{p}\mathbf{p}^\top)\alpha(\mathbf{e}_k - \mathbf{p})$$
$$= \alpha [\mathbf{p} \odot (\mathbf{e}_k - \mathbf{p}) - \mathbf{p}(p_k - \|\mathbf{p}\|_2^2)],$$

and

$$\delta p_i = \alpha [p_i(\mathbf{1}\{i = k\} - p_i) - p_i(p_k - \|\mathbf{p}\|_2^2)].$$

As the first-order entropy change is given in (5), we substitute $\delta p_i$ and apply $\sum_i p_i \log p_i = -H$ to (5), which yields:

$$\Delta H = \alpha \left[\sum_i p_i^2(H + \log p_i) - p_k(H + \log p_k)\right] + O(\alpha^2)$$
$$= -\alpha [S_* - \mathbb{E}_{i\sim\mathbf{p}}[S_i]] + O(\alpha^2).$$

The proof is completed by applying the definition of $S_i$ and $\mathbb{E}_{i\sim\mathbf{p}}[S_i]$. □

**Implications** Theorem 3.3 reveals a crucial distinction from the single-logit case. With a GRPO optimization step, the entropy change is no longer governed by the absolute value of the entropy discriminant score $S_*$, but by its deviation from the policy-weighted expectation $\mathbb{E}_{i\sim\mathbf{p}}[S_i]$, which acts as a dynamic baseline. Entropy decreases if we reward (positive $\alpha$) a token $a^k$ whose score $S_*$ is above the baseline, and it increases when $S_*$ is below the baseline. The relationship can be reversed when we penalize (negative $\alpha$) a token. Since $\alpha = \eta r A$, with $\eta$ on the order of $10^{-6}$, $r$ clipped near 1, and $A$ typically $O(1)$ due to within-group standardization, we have $|\alpha| \ll 1$. Therefore, the $O(\alpha^2)$ terms are negligible and the first-order approximation in Theorem 3.3 is accurate in practice.

Moving a step forward, we provide two corollaries derived from Theorem 3.3.

**Corollary 3.4.** *To a first-order approximation, with on-policy sampling, the expected entropy change factor $S_k - \mathbb{E}_{i\sim p}[S_i]$ of a token within GRPO optimization is zero, i.e.,*

$$\mathbb{E}_{k\sim\mathbf{p}}[S_* - \mathbb{E}_{i\sim\mathbf{p}}[S_i]] = 0. \tag{12}$$

**Corollary 3.5.** *For on-policy GRPO training with a batch, the expected value of entropy change factor $S_*^t - \mathbb{E}_{i\sim\mathbf{p}_t}[S_i^t]$ over the batch of tokens $\mathcal{T}_\mathcal{B}$ is zero:*

$$\mathbb{E}_{t\in\mathcal{T}_\mathcal{B}}[S_*^t - \mathbb{E}_{i\sim\mathbf{p}_t}[S_i^t]] = 0. \tag{13}$$

We provide the proofs for these two corollaries in Appendix A. These corollaries demonstrate that, from both the vocabulary and batch perspectives, the discriminant score $S_*$ possesses a favorable decentralization property under on-policy sampling. Therefore, imposing constraints on tokens based on the value of $S_*$ relative to its expectation offers a simple and direct approach to regulating entropy dynamics. Based on such analysis, we propose two methods for constraining entropy in Section 4.1.

Considering the potential distribution of the advantage term under model sampling or the statistical distribution across multiple tokens in a training batch, these two corollaries can be further extended to incorporate the complete expectation, including the advantage term. Detailed discussions are provided in Appendix C, which help further understand the causes of entropy collapse in RFT.

# 4. Bridging Entropy Dynamics to Entropy Control Methods

## 4.1. Entropy Discriminator Guided Clipping

The theoretical analysis provides novel insights into the relationships between the discriminator score $S_*^t$ and the entropy dynamics in RFT. Upon this, we can effectively identify tokens within a training batch that exert a disproportionate impact on entropy changes, enabling us to selectively mitigate the influence of such outlier tokens for achieving fine-grained and flexible control over the entropy regularization throughout the training process.

Inspired by Theorem 3.2, we propose a simple yet effective batch-level clipping method.

**Algorithm 4.1.** *(Clip$_\mathcal{B}$: Batch-Normalized Entropy-Discriminator Clipping):*

*Let $\mathcal{T}_\mathcal{B}$ denote the set of all tokens in the responses of a given batch $\mathcal{B}$. We first compute the batch-wise mean of the discriminator scores, $\bar{S} = \mathbb{E}_{t \in \mathcal{T}_\mathcal{B}}[S_*^t]$, and the corresponding standard deviation, $\sigma = \sqrt{\mathbf{Var}_{t \in \mathcal{T}_\mathcal{B}}[S_*^t]}$. During the RFT process, we only preserve the gradients associated with those tokens that satisfy a specific condition by applying the following mask $m_t$ to each token $t$:*

$$m_t = \mathbf{1}\left\{-\mu^- \sigma \leq S_*^t - \bar{S} \leq \mu^+ \sigma\right\}. \tag{14}$$

Here $\mu^+$ and $\mu^-$ are used to control the clipping threshold based on the degree of outlierness. This algorithm identifies the effects of each token on entropy change and filters out tokens that contribute to severe fluctuations in $\Delta H$. This is accomplished by simply examining the logits of response tokens, combined with the proposed batch-level normalization. This operation requires minimal computation on scalar values rather than high-dimensional tensors, thus introducing negligible additional computational cost and allowing for easy integration into existing training frameworks.

Moreover, Theorem 3.3 provides a more precise characterization of the entropy change, particularly in the context of GRPO. This analysis motivates us to derive the vocabulary-normalized entropy-discriminator clipping method.

**Algorithm 4.2.** *(Clip$_\mathcal{V}$: Vocabulary-Normalized Entropy-Discriminator Clipping):*

*For each token $t$ in a batch $\mathcal{T}_\mathcal{B}$, we first define its vocabulary-centered score as $S_c^t = S_*^t - \mathbb{E}_{i \sim \mathbf{p}_t}[S_t^i]$, where $\mathbf{p}_t$ is the policy's predictive distribution over the vocabulary $\mathcal{V}$ at step $t$. We then compute the standard deviation of these centered scores in a batch: $\sigma' = \sqrt{\mathbf{Var}_{t \in \mathcal{T}_\mathcal{B}}[S_c^t]}$. As established in Corollary 3.5, the batch-mean of these centered scores, $\mathbb{E}_{t \in \mathcal{T}_\mathcal{B}}[S_c^t]$, approximates zero. This simplifies the clipping condition. The mask for a token $t$ is thus defined as:*

$$m_t = \mathbf{1}\left\{-\mu^- \sigma' \leq S_*^t - \mathbb{E}_{i \sim \mathbf{p}_t}[S_i^t] \leq \mu^+ \sigma'\right\}. \tag{15}$$

In Clip$_\mathcal{V}$, the computation of $\mathbb{E}_{i \sim \mathbf{p}_t}[S_i^t]$ introduces some computational overhead. Fortunately, the quantities required to compute this term, such as the policy's logits over the full vocabulary, are often available as intermediate results from the forward pass used for entropy and log-probability calculations. This allows us to evaluate the expectation at a relatively low additional cost.

In Section 5.3, we provide empirical studies on the effectiveness of Clip$_\mathcal{B}$ and Clip$_\mathcal{V}$.

## 4.2. Interpreting Existing Methods through Entropy Dynamics

Recent works have proposed various entropy-based methods to enhance training stability and effectiveness. These methods, however, are often developed from heuristic principles and can necessitate labor-intensive hyperparameter tuning without clear theoretical guidance. To provide a better understanding of their underlying mechanisms, we re-examine these methods through the lens of our entropy dynamics analysis (refer to Section 3).

We categorize the related studies into three groups: (i) *Clipping Mechanisms*, which stabilize the optimization by constraining the updates of token probability. Representative works include the clipping operation in GRPO (Guo et al., 2025), the clip-higher method in DAPO (Yu et al., 2025) and the separate clipping mechanism in CE-GPPO (Su et al., 2025). (ii) *Entropy Regularization*, which regularizes updates to tokens with high entropy, as proposed by Wang et al. (2025). (iii) *Probability Weighted Updating*, which constrains the updates based on token probabilities, exemplified by methods from (He et al., 2025; Yang et al., 2025b).

Before we conduct the investigation and interpretation, let's first recall Theorem 3.3 and examine, from a statistical perspective, the relationship it reveals between two factors often considered by the methods above, i.e., probability and entropy. The first term in Theorem 3.3, $S_* = p_k(H + \log p_k)$, directly associates these two factors. For tokens sampled with high probability, $S_*$ tends to be larger; similarly, tokens sampled in positions with high entropy also have larger $S_*$. Tokens sampled with larger $S_*$ are more likely to obtain a positive value when calculating the deviation from the expectation $\mathbb{E}_{i \sim \mathbf{p}}[S_i]$, as the expectation represents an average value under the current model's sampling distribution.

As a result, when considering entropy change, for positive samples, higher probability and lower token entropy are often associated with a decrease in entropy. In contrast, lower probability and higher entropy are often linked to an increase in entropy. For negative samples, the trend reverses.

**Clipping Mechanisms** Clipping in GRPO can be formu-

lated as a gradient mask for the $t$-th token in response $i$:

$$M_{i,t} = \mathbf{1}\{A_i > 0, r_{i,t}(\theta) \leq 1 + \epsilon_{\text{high}}\}$$
$$+ \mathbf{1}\{A_i < 0, r_{i,t}(\theta) \geq 1 - \epsilon_{\text{low}}\}, \quad (16)$$

where $r_{i,t}(\theta) = \frac{\pi_\theta(o_{i,t}|q,o_{i,<t})}{\pi_{\text{sample}}(o_{i,t}|q,o_{i,<t})}$ is the importance ratio. The clipping mechanism prevents an excessive increase in probability for tokens in positive samples and an excessive decrease for tokens in negative samples. Due to the nature of the importance ratio, this mechanism predominantly affects tokens with low probabilities under the sampling policy.

Empirical statistics from (Yu et al., 2025) show that clipped tokens typically have a maximum probability of around 0.15. Across a training trajectory where overall token entropy declines. As we analyzed above, these low-probability tokens are associated with the condition $S_* - \mathbb{E}[S_i] < 0$. For these tokens, the sign of the entropy change is given by $\text{sign}(\Delta H) = -\text{sign}(\varepsilon)\text{sign}(S_* - \mathbb{E}_{i\sim\mathbf{p}}[S_i]) = \text{sign}(\varepsilon)$. Consequently, updates on positive samples ($\text{sign}(\varepsilon) = +1$) tend to increase token entropy, while updates on negative samples ($\text{sign}(\varepsilon) = -1$) tend to decrease it. The overall entropy dynamics is a superposition of these two effects: in most cases, it manifests as a rapid decline in entropy, while in others, it exhibits complex fluctuations (Liu et al., 2025).

The clip-higher method in DAPO, which sets a larger $\epsilon_{\text{high}}$ for positive samples, corresponds to this insight. By relaxing the clipping constraint for positive samples, it preserves their entropy-increasing updates. This targeted intervention counteracts the tendency of entropy decrease during RFT, thereby promoting more exploration and better performance.

Consistent with our theoretical framework, Su et al. (2025) empirically demonstrates that high-probability positive tokens and low-probability negative tokens tend to suppress exploration, whereas low-probability positive tokens and high-probability negative tokens encourage exploration.

**Entropy Regularization** Entropy regularization refers to methods that compute gradients only for a certain proportion of tokens with high entropy. For example, Wang et al. (2025) demonstrates improved performance by applying updates to only the top 20% of tokens with the highest entropy.

As we analyzed above, high token entropy corresponds to a condition where our theoretical quantity $S_* - \mathbb{E}_{i\sim\mathbf{p}}[S_i]$ is likely to be positive. According to Theorem 3.3, for these tokens, the updates on positive samples would decrease entropy, while updates on negative samples would increase it. The net effect on entropy is therefore determined by the balance between these two opposing forces. The empirical results in (Wang et al., 2025), which show that as the proportion of selected high-entropy tokens is varied, the overall entropy first increases and then decreases relative to a baseline, provide strong evidence for this trade-off.

**Probability Weighted Updating** Similar to entropy regularization, Probability Weighted Updating methods constrain or scale token updates based on their probabilities. For example, He et al. (2025) proposes to assign higher weights to positive samples with low probability. In the context of our analysis, low-probability tokens are associated with $S_* - \mathbb{E}_{i\sim\mathbf{p}}[S_i] < 0$. When these tokens are part of a positive sample, the expected change in entropy is positive. By amplifying the updates for this specific subset of tokens, the method explicitly promotes gradients that increase token entropy, alleviating the entropy collapse issue. The provided experimental results support this conclusion.

In summary, our analysis offers a unified view for understanding the mechanics of existing methods, which function by amplifying the effects of tokens contributing to entropy increase or suppressing those leading to entropy decrease, thereby preventing entropy collapse in RFT.

## 5. Experiments

### 5.1. Settings

We select the Qwen2.5-7B-Instruct and Qwen2.5-14B-Instruct (Yang et al., 2024) as our base models for RFT, utilizing the DAPO-Math-17k dataset (Yu et al., 2025) as our training set. Following previous studies (Lightman et al., 2023), we exclude 500 questions from the training set to form the validation set (denoted by DAPO500). We filter out samples from the training set with excessively high ($\geq 15/16$) or low ($\leq 1/16$) pass rates, as evaluated by Qwen2.5-7B-Instruct.

For evaluation, we adopt two challenging mathematical datasets, i.e., AIME24, AIME25, AMC23 and Minerva Math (Lewkowycz et al., 2022) to form our test set. We adopt the Avg@32/Pass@32 evaluation metrics for AIME24, AIME25 and AMC23 and Avg@8/Pass@8 for Minerva and DAPO500. Here Avg@K denotes the average accuracy across K responses for each question, while Pass@K represents the probability that at least one of K responses is correct.

### 5.2. Empirical Observations of the Entropy Dynamics

We first provide empirical evidence supporting Theorem 3.2, which posits a close relationship between the discriminator score $S_*$ and the direction of change in token entropy, i.e., $\text{sign}(\Delta H)$. Specifically, during the training process, we selectively update the loss associated with tokens exhibiting $S_* > 0$ or $S_* < 0$. The standard training process serves as our baseline for comparison. For clear observations, we apply these selective updates to positive (rewarding) and negative (punishing) samples separately, presenting the results in Figures 1(a) and 1(b), respectively. These results align with our analysis. For example, in Figure 1(a), when

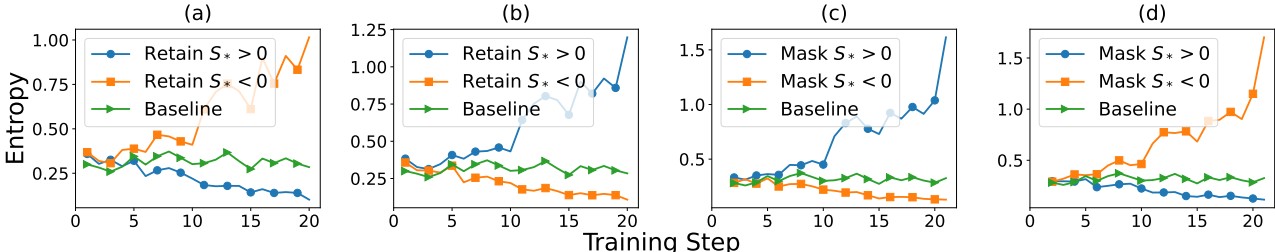

*Figure 1.* We retain or mask the gradients of tokens satisfying $S_* > 0$ or $S_* < 0$, respectively. The resulting entropy changes are shown in (a,c) for positive samples, and (b,d) for negative samples.

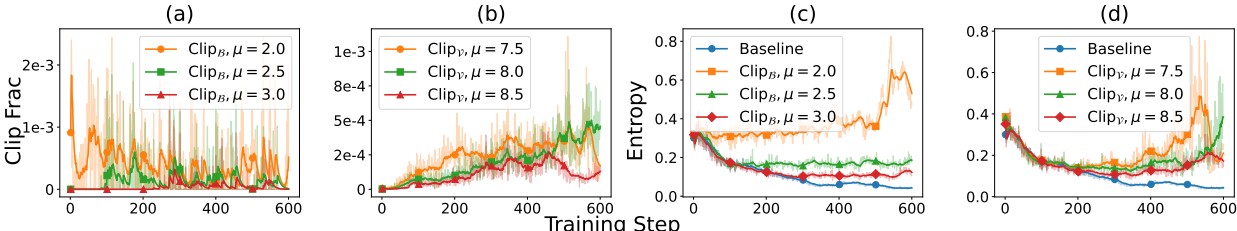

*Figure 2.* The effects of $\text{Clip}_\mathcal{B}$ and $\text{Clip}_\mathcal{V}$ with different $\mu$ in controlling clip fraction and entropy.

we only retain updates for tokens with $S_* > 0$ in positive samples, we observe a decrease in entropy consistent with $\text{sign}(\Delta H) = -\text{sign}(\varepsilon) \cdot \text{sign}(S_*) < 0$. Conversely, retaining updates for tokens with $S_* < 0$ induces an increment in entropy. Such a phenomenon is precisely reversed in Figure 1(b) as we apply these operations to negative samples.

To further probe this relationship, we investigate a practical scenario where we mask the gradients of tokens that satisfy specific conditions during the training process. Similarly, as shown in Figure 1(c), when the gradients associated with tokens in positive samples that satisfy $S_* > 0$ are masked (these are believed to contribute to entropy decrease), the entropy increases uncontrollably. Conversely, masking the gradients of tokens that satisfy $S_* < 0$ leads to a continuous decrease in entropy. Figure 1(d) illustrates that performing the same masking operations on negative samples results in the opposite behavior. These experimental results further confirm our analysis, which suggests that the sign of the discriminator score $S_*$ is a reliable predictor of tokens' influence on entropy dynamics within RFT.

In Figure 3, we illustrate the distribution of $S_*$ within a training batch and its deviation from its sampling expectation, as involved in Theorem 3.3. The value of $\mathbb{E}_{t \sim \mathcal{T}_\mathcal{B}}[S_*^t - \mathbb{E}_{i \sim \mathbf{p}}[S_i^t]]$ is three orders of magnitude smaller than that of $\mathbb{E}_{t \sim \mathcal{T}_\mathcal{B}}[S_*^t]$, and approaches zero, effectively validating Corollary 3.5.

### 5.3. Effects of Entropy Discriminator Clipping Methods

In this subsection, we validate the effectiveness of the clipping methods proposed in Section 4.1, including $\text{Clip}_\mathcal{B}$

and $\text{Clip}_\mathcal{V}$. Considering that entropy empirically exhibits a clear decreasing trend within RFT, we choose negative samples as the primary focus and apply our clipping methods to mask the losses of specific tokens. As shown in Figures 2(a) and 2(b), the hyper-parameter $\mu$ in the $\text{Clip}_\mathcal{B}$ and $\text{Clip}_\mathcal{V}$ methods provides effective control over the number of clipped tokens (a larger $\mu$ indicates a smaller clip proportion), thereby supporting flexible adjustment of the intervention intensity. In Figures 2(c) and 2(d), we illustrate the effects of the clipping methods in controlling entropy with different values of $\mu$. We observe that both $\text{Clip}_\mathcal{B}$ and $\text{Clip}_\mathcal{V}$ successfully mitigate the entropy decay to excessively low levels, as in the baseline (the standard RFT training).

Existing RFT studies (Yu et al., 2025; Liao et al., 2025) suggest that maintaining a certain level of entropy can retain the model's exploration capabilities, leading to better model performance. Therefore, we validate the performance of models trained using $\text{Clip}_\mathcal{B}$ and $\text{Clip}_\mathcal{V}$, as summarized in Table 1. These results demonstrate that both $\text{Clip}_\mathcal{B}$ and $\text{Clip}_\mathcal{V}$ achieve outperformance compared to standard GRPO across various datasets, and is competitive to other baselines. The results confirm their effect in preserving model exploration by controlling entropy.

Besides, we also extend the evaluation across two dimensions: the training algorithm (by integrating with PPO) and model diversity (including Qwen3-4B-Base, DeepSeek-Distilled-Llama3-8B, and InternLM3-8B). As shown in Appendix E, the consistent performance gains across these settings confirm that our methods are effective at preserving model exploration and enhancing model performance.

*Table 1.* Avg@K performance of baselines and our methods on virous datasets. Best and second-best results are marked within each model block. Avg is the arithmetic average over AIME24, AIME25, AMC23, Minerva, and DAPO500.

| Base Model | Method | AIME24 | AIME25 | AMC23 | Minerva | DAPO500 | Avg |
|---|---|---|---|---|---|---|---|
| Qwen2.5-7B | Vanilla GRPO | 16.88 | 15.42 | 61.64 | 54.83 | 48.03 | 39.36 |
| | Clip-Cov (Cui et al., 2025) | 16.67 | 15.00 | 62.11 | 55.11 | 47.98 | 39.37 |
| | KL-Cov (Cui et al., 2025) | 17.40 | **16.77** | 60.39 | 56.53 | 46.60 | 39.54 |
| | Rewarding Unlikely (He et al., 2025) | 15.94 | 15.83 | 56.25 | 57.67 | 48.22 | 38.78 |
| | Forking Tokens (Wang et al., 2025) | 16.36 | 14.90 | 59.30 | 53.98 | 47.13 | 38.33 |
| | GRPO+Clip$_\mathcal{B}$ | **19.69** | 16.36 | **62.66** | **58.52** | **49.68** | **41.38** |
| | GRPO+Clip$_\mathcal{V}$ | 18.12 | 15.94 | 61.95 | 57.39 | 49.65 | 40.61 |
| Qwen2.5-14B | Vanilla GRPO | 22.50 | 17.60 | 66.33 | 68.47 | 52.95 | 45.57 |
| | Clip-Cov (Cui et al., 2025) | 22.97 | 20.62 | **70.31** | 69.03 | 59.85 | 48.56 |
| | KL-Cov (Cui et al., 2025) | 22.19 | 19.95 | 69.53 | 70.45 | 59.05 | 48.23 |
| | Forking Tokens (Wang et al., 2025) | 21.15 | 20.42 | 67.58 | 69.60 | 58.15 | 47.38 |
| | GRPO+Clip$_\mathcal{B}$ | 23.33 | 20.62 | 70.16 | 69.60 | 60.35 | 48.81 |
| | GRPO+Clip$_\mathcal{V}$ | **23.44** | **21.35** | 69.45 | **71.02** | **61.92** | **49.44** |

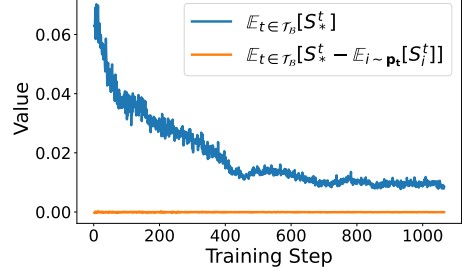

*Figure 3.* The batch-averaged value of $S_*$ and $S_* - \mathbb{E}_{i \sim \mathbf{p}}[S_i]$.

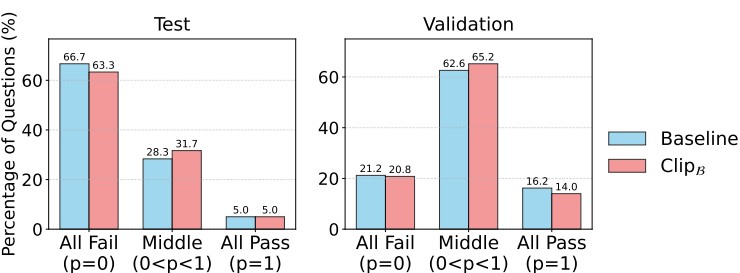

*Figure 4.* Comparison between Clip$_\mathcal{B}$ and vanilla GRPO on the distribution of problem pass rates.

## 5.4. Analysis of Exploration versus Exploitation

In Table 2, we compare model performance using Pass@K and Avg@K. A significant gain in Pass@K indicates that the model can generate diverse responses for solving problems (exploration), whereas improvements in Avg@K primarily reflect exploitation of similar high-reward patterns.

The experimental results demonstrate that our methods not only achieve significant improvements in both Pass@K and Avg@K across all datasets. These results confirm that stabilizing entropy with the proposed clipping method fosters greater solution diversity and encourages the model to discover correct reasoning paths for a wider array of problems, instead of merely sharpening confidence on those easier ones.

Moreover, we further conduct a study on the distribution of pass rates among multiple rollouts for individual problems. Taking the Qwen-2.5-7B-Instruct model and the Clip$_\mathcal{B}$ method as an example, we illustrate the results in Figure 4. For the standard GRPO, the proportion of problems that

are completely solved or completely failed is significantly higher than that of Clip$_\mathcal{B}$. This indicates that GRPO excessively prioritizes exploitation while neglecting the importance of exploration. Conversely, Clip$_\mathcal{B}$ focuses more on encouraging exploration, resulting in a pass rate distribution that is more concentrated around the middle range. This further confirms that the performance gains achieved by our method stem from encouraging the model to explore solutions for a broader range of problems, rather than simply memorizing easier problems that could be solved with higher certainty.

## 6. Related Works

Reinforcement fine-tuning (RFT) has been widely adopted for tuning LLMs, including representative methods such as GRPO (Shao et al., 2024), DAPO (Yu et al., 2025), and GSPO (Zheng et al., 2025).

Early studies (Chen et al., 2026; Wu et al., 2025) observed that, during RFT, the exploratory capacity of LLMs gradually decrease, a phenomenon typically manifested as en-

*Table 2.* Comparison of vanilla GRPO and our methods on Avg@K and Pass@K ($K$=32 for AIME24/25 and $K$=8 for DAPO500).

| Method | AIME24 | | AIME25 | | DAPO500 | |
|---|---|---|---|---|---|---|
| | Avg@K | Pass@K | Avg@K | Pass@K | Avg@K | Pass@K |
| Qwen2.5-7B-Inst | 11.35 | 36.67 | 6.67 | 33.33 | 31.55 | 70.2 |
| GRPO | 16.88 | 50.00 | 15.42 | 50.00 | 48.03 | 76.8 |
| GRPO+Clip$_{\mathcal{B}}$ | **19.69** (+2.81) | **56.67** (+6.67) | **16.35** (+0.93) | 53.33 (+3.33) | **49.68** (+1.65) | **80.2** (+3.4) |
| GRPO+Clip$_{\mathcal{V}}$ | 18.12 (+1.24) | 53.33 (+3.33) | 15.94 (+0.52) | **56.67** (+6.67) | 49.65 (+1.62) | 79.0 (+2.2) |
| Qwen2.5-14B-Inst | 12.14 | 41.67 | 11.72 | 38.33 | 40.22 | 74.7 |
| GRPO | 22.50 | 66.33 | 17.60 | 50.00 | 52.95 | 84.0 |
| GRPO+Clip$_{\mathcal{B}}$ | 23.33 (+0.83) | **66.67** (+0.34) | 20.62 (+3.02) | **56.67** (+6.67) | 60.35 (+7.40) | 85.6 (+1.6) |
| GRPO+Clip$_{\mathcal{V}}$ | **23.44** (+0.94) | **66.67** (+0.34) | **21.35** (+3.75) | **56.67** (+6.67) | **61.92** (+8.97) | **86.6** (+2.6) |

tropy collapse (Yu et al., 2025). To better understand this issue, several studies have examined the relationship between token entropy and reasoning performance in RFT. For instance, He et al. (Wang et al., 2025) shows that the top 20% high-entropy tokens serve as reasoning forking tokens and therefore carry greater training value, while Qian et al. (Qian et al., 2026) further demonstrates from an information-theoretic perspective that a small number of thinking tokens can exert a disproportionately large influence on reasoning performance. Jin et al. (Jin et al., 2025) further investigates why preserving entropy is beneficial for RFT training.

In parallel, another line of work has focused on alleviating entropy collapse through various practical techniques, thereby improving training effectiveness. On the one hand, some studies introduce explicit entropy regularization (Hu et al., 2025; Jiang et al., 2025; Cheng et al., 2026; Shen, 2025), or more directly optimize Pass@K accuracy as the training objective (Chen et al., 2025b). On the other hand, some methods attempt to address the underlying causes of entropy collapse based on specific empirical or theoretical insights, for example through flexible clipping schemes (Yu et al., 2025; Su et al., 2025) or sample weighting based on token probabilities (He et al., 2025; Yang et al., 2025b). In addition, Liu et al. (Liu et al., 2025) systematically evaluates the effectiveness of many such techniques across different models and provides practical guidance for their use.

Most closely related to our work is a third line of research that investigates how different tokens contribute to entropy dynamics. Cui et al. (Cui et al., 2025) connects entropy changes to model sampling distributions and models the relationship between performance and entropy. While establishing a solid theoretical foundation, it relies on the advantage of unsampled tokens, which is difficult to estimate in most RFT algorithms. Hao et al. (Hao et al., 2025) and Xi et al. (Xi et al., 2025) propose that positive and negative sample tokens can be partitioned into four regions

according to their probability ranges, an observation broadly consistent with our analysis. Petrenko et al. (Petrenko et al., 2026) investigates the patterns of entropy change of different RFT algorithms and offers a number of useful insights and tricks from an empirical perspective.

## 7. Conclusions

In this study, we focus on a theoretical framework to provide a principled understanding of entropy dynamics in RFT. We quantify the entropy change and further extend this analysis to a practical GRPO optimization step, revealing that entropy fluctuations arise from the combined effect of a token's update direction, its probability, and the policy entropy. These insights offer explanations for the commonly observed entropy collapse phenomenon, guide the development of entropy controlling strategies, and unify the interpretation of existing entropy-based methods. We hope that the theoretical framework can foster a clear understanding of the underlying mechanisms of entropy dynamics in RFT, thereby accelerating progress in the field.

## 8. Limitations

Our framewrok explains the evolution of token entropy during the RFT process. However, in the practical training of black-box neural networks, different tokens may influence each other. Additionally, the principled relationship between entropy and model performance remains to be further investigated. Finally, we look forward to future work extending our framework to a broader range of domains.

## Impact Statement

This paper presents work whose goal is to advance the field of machine learning. There are many potential societal consequences of our work, none of which we feel must be specifically highlighted here.

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

# A. Proof of Corollaries

**Corollary 3.4.** *To a first-order approximation, with on-policy sampling, the expected entropy change of a token within GRPO optimization is zero, i.e.*

$$\mathbb{E}_{k\sim\mathbf{p}}\left[S_* - \mathbb{E}_{i\sim\mathbf{p}}[S^i]\right] = 0.$$

*Proof.* We derive the results as follows:

$$
\begin{aligned}
\mathbb{E}_{k\sim\mathbf{p}}\left[S_* - \mathbb{E}_{i\sim\mathbf{p}}[S^i]\right] &= \mathbb{E}_{k\sim\mathbf{p}}\left[p_k\big(H + \log p_k\big) - \sum_{i=1}^{V} p_i^2\big(H + \log p_i\big)\right] \\
&= \sum_{k=1}^{V} p_k^2\big(H + \log p_k\big) - \sum_{i=1}^{V} p_i^2\big(H + \log p_i\big) \\
&= 0.
\end{aligned}
$$

$\square$

**Corollary 3.5.** *For on-policy GRPO training with a batch, the expected value of entropy change factor $S_*^t - \mathbb{E}_{i\sim\mathbf{p}_t}[S_i^t]$ over the batch of tokens $\mathcal{T}_\mathcal{B}$ is zero:*

$$\mathbb{E}_{t\in\mathcal{T}_\mathcal{B}}\left[S_*^t - \mathbb{E}_{i\sim\mathbf{p}_t}[S_i^t]\right] = 0. \tag{17}$$

*Proof.* For each token $t \in \mathcal{T}_\mathcal{B}$, let $\mathbf{p}_t = (p_t^1, \ldots, p_t^V)$ be the on-policy token distribution, $H_t = -\sum_{i=1}^{V} p_t^i \log p_t^i$, and $S_t^i := p_t^i\big(H_t + \log p_t^i\big)$. Draw the action index on-policy: $K_t \sim \mathrm{Cat}\big(\mathbf{p}_t\big)$. Then, conditioning on $\mathbf{p}_t$,

$$\mathbb{E}\left[S_t^{K_t}\,\Big|\,\mathbf{p}_t\right] = \sum_{i=1}^{V} p_t^i\, S_t^i = \sum_{i=1}^{V} (p_t^i)^2\big(H_t + \log p_t^i\big) = \mathbb{E}_{i\sim\mathbf{p}_t}[S_t^i].$$

Hence $\mathbb{E}\left[S_t^{K_t} - \mathbb{E}_{i\sim\mathbf{p}_t}[S_t^i]\,\Big|\,\mathbf{p}_t\right] = 0$ for each token $t$. Averaging over the batch and using linearity of expectation,

$$\mathbb{E}\left[\frac{1}{|\mathcal{T}_\mathcal{B}|}\sum_{t\in\mathcal{T}_\mathcal{B}}\Big(S_t^{K_t} - \mathbb{E}_{i\sim\mathbf{p}_t}[S_t^i]\Big)\,\bigg|\,\{\mathbf{p}_t\}_{t\in\mathcal{T}_\mathcal{B}}\right] = \frac{1}{|\mathcal{T}_\mathcal{B}|}\sum_{t\in\mathcal{T}_\mathcal{B}} 0 = 0.$$

Finally, applying the tower property removes the conditioning and yields the stated result. $\square$

# B. Detailed Experiment Setup

All experiments are conducted on NVIDIA A100 and H20 GPUs. We implement the experiment with the Trinity-RFT (Pan et al., 2025) framework.

For the training process, we adopt the Adam optimizer with hyperparameters $(0.9, 0.999)$. We set the training batch size to 64, the number of rollouts to 16 for Qwen2.5-7B-Instruct and Qwen2.5-14B-Instruct and 8 for other models, and employ a learning rate of $4 \times 10^{-7}$. The temperature is set to 1.0 for sampling rollouts and 0.7 for evaluation.

**Reward design** The reward for each response is determined by its answer correctness, i.e., (1) $r_i = 1$ if the answer and format are correct; (2) $r_i = 0$, otherwise.

# C. Extension to Advantage-Aware Analysis

In this section, we extend the findings of Corollary 3.4 and Corollary 3.5 to incorporate advantage estimation. We analyze the expectations of the full entropy change expression presented in Theorem 3.3 from two complementary perspectives: model sampling and batch averaging.

## C.1. Extension of Corollary 3.4 to Model Sampling

We assume that at each position $t$, every token ID in the vocabulary has a latent advantage value, i.e., $A = \mathbf{A}(i)$ for $i \sim \mathbf{p}_t$. Building upon Theorem 3.3, we derive the following corollary regarding the expected entropy change.

**Corollary C.1.** *For on-policy GRPO training, the first-order expectation of the token-wise entropy change is given by:*

$$\mathbb{E}_{k \sim \mathbf{p}}[\Delta H] = -\eta \operatorname{Cov}_{k \sim \mathbf{p}}(A, S_* - \mathbb{E}_{i \sim \mathbf{p}}[S_i]). \tag{18}$$

*Proof.* We begin by applying the result from Theorem 3.3. Recall that $\alpha = \eta r A$. Considering the constant $\eta$ and $r = 1$ in the on-policy setting, the first-order expectation of token entropy change under on-policy sampling can be given by:

$$\mathbb{E}_{k \sim \mathbf{p}}[\Delta H] = -\eta \mathbb{E}_{k \sim \mathbf{p}}\big[A(S_* - \mathbb{E}_{i \sim \mathbf{p}}[S_i])\big]. \tag{19}$$

We decompose the covariance in this equation, which gives:

$$\begin{aligned}
\mathbb{E}_{k \sim \mathbf{p}}[\Delta H] = -\,\eta\big\{ &\mathbb{E}_{k \sim \mathbf{p}}[A]\mathbb{E}_{k \sim \mathbf{p}}[S_*] + \operatorname{Cov}_{k \sim \mathbf{p}}(A, S_*) \\
&- \mathbb{E}_{k \sim \mathbf{p}}[A]\mathbb{E}_{k \sim \mathbf{p}}[\mathbb{E}_{i \sim \mathbf{p}}[S_i]] - \operatorname{Cov}_{k \sim \mathbf{p}}(A, \mathbb{E}_{i \sim \mathbf{p}}[S_i])\big\}.
\end{aligned}$$

By Corollary 3.4, we have $\mathbb{E}_{k \sim \mathbf{p}}[S_*] - \mathbb{E}_{k \sim \mathbf{p}}[\mathbb{E}_{i \sim \mathbf{p}}[S_i]] = 0$. Substituting this into the equation above, we have:

$$\mathbb{E}_{k \sim \mathbf{p}}[\Delta H] = -\eta \operatorname{Cov}_{k \sim \mathbf{p}}(A, S_* - \mathbb{E}_{i \sim \mathbf{p}}[S_i]).$$

$\square$

**Implications.** Corollary C.1, based on the on-policy policy gradient formula, provides a clean expression for entropy change. It decouples the advantage from the core entropy change term $S_* - \mathbb{E}_{i \sim \mathbf{p}}[S_i]$, and establishes their relationship through a covariance.

It is worth noting that, In GRPO, the advantage for tokens not actually sampled is undefined and non-computable; therefore, Corollary C.1 cannot be directly applied in algorithmic implementation. Nevertheless, it offers a theoretical potential for understanding entropy collapse in the GRPO training process.

In GRPO, the way advantages are obtained is coupled with the policy model distribution, which promotes entropy collapse. We will verify this hypothesis from a batch-level perspective in the next subsection.

## C.2. Extension of Corollary 3.5

The batch-level entropy is defined by the arithmetic average of token entropies within a batch:

$$H_{\mathcal{T}_\mathcal{B}} = \frac{1}{|\mathcal{T}_\mathcal{B}|} \sum_{t \in \mathcal{T}_\mathcal{B}} H_t$$

Therefore, batch-level entropy change is also the arithmetic average of token entropy change:

$$\Delta H_{\mathcal{T}_\mathcal{B}} = \frac{1}{|\mathcal{T}_\mathcal{B}|} \sum_{t \in \mathcal{T}_\mathcal{B}} \Delta H_t \tag{20}$$

Based on the above definition, we derive the following corollary:

**Corollary C.2.** *For on-policy GRPO training with a batch, the first-order batch-wise entropy change of tokens $\mathcal{T}_\mathcal{B}$ is given by:*

$$\Delta H_{\mathcal{T}_\mathcal{B}} = -\eta \operatorname{Cov}_{\mathcal{B}}(A, S_* - \mathbb{E}_{i \sim \mathbf{p}}[S_i]). \tag{21}$$

*Proof.* Applying the results in Theorem 3.3 into equation 20 gives:

$$\Delta H_{\mathcal{T}_\mathcal{B}} = -\frac{1}{|\mathcal{T}_\mathcal{B}|} \sum_{t \in \mathcal{T}_\mathcal{B}} \alpha(S_* - \mathbb{E}_{i \sim \mathbf{p}_t}[S_i^t]) = -\mathbb{E}_\mathcal{B}[\alpha(S_* - \mathbb{E}_{i \sim \mathbf{p}}[S_i])], \tag{22}$$

where $\mathbb{E}_{\mathcal{B}}$ refers to statistical expectation (i.e., an arithmetic average over batch $\mathcal{B}$).

Recall the definition of $\alpha = \eta r A$: the learning rate $\eta$ is constant within a batch; $r$ is constantly 1 in the on-policy setting. The advantage $A$ estimated in the GRPO algorithm is NOT independent of the chosen token id in a batch $\mathcal{T}_{\mathcal{B}}$, i.e.,

$$\Delta H_{\mathcal{T}_{\mathcal{B}}} = -\eta\, \mathbb{E}_{\mathcal{B}}[A(S_* - \mathbb{E}_{i\sim\mathbf{p}}[S_i])].$$

We further apply covariance decomposition to $\Delta H_{\mathcal{T}_{\mathcal{B}}}$ within a training batch:

$$\Delta H_{\mathcal{T}_{\mathcal{B}}}/\eta = -\{\mathbb{E}_{\mathcal{B}}[A]\mathbb{E}_{\mathcal{B}}[S_*] + \mathrm{Cov}_{\mathcal{B}}(A, S_*) - \mathbb{E}_{\mathcal{B}}[A]\mathbb{E}_{\mathcal{B}}[\mathbb{E}_{i\sim\mathbf{p}}[S_i]] - \mathrm{Cov}_{\mathcal{B}}(A, \mathbb{E}_{i\sim\mathbf{p}}[S_i])\}.$$

According to Corollary 3.5, we have $\mathbb{E}_{\mathcal{B}}[S] - \mathbb{E}_{\mathcal{B}}[\mathbb{E}_{i\sim\mathbf{p}}[S_i]] = 0$, which gives:

$$\Delta H_{\mathcal{T}_{\mathcal{B}}}/\eta = -\mathrm{Cov}_{\mathcal{B}}(A, S_* - \mathbb{E}_{i\sim\mathbf{p}}[S_i]). \tag{23}$$

Finally, we multiply both sides of the above equation by $\eta$ to complete the proof. $\square$

**Implications.** Corollary C.2 provides a computable form analogous to Corollary C.1 from the batch perspective. We conduct a experiment to monitored the quantity $-\mathrm{Cov}_{\mathcal{B}}(A, S_* - \mathbb{E}_{i\sim\mathbf{p}}[S_i])$ during training. As shown in Figure 5, its value has a larger magnitude in the negative portion compared with the positive ones. This observation further validates the hypothesis in Appendix C.1. The model tends to obtain correct answers (i.e., $A > 0$) by producing "safe" responses, those with relatively high probability, for which $S_* - \mathbb{E}[S_i]$ tends to be positive, whereas exploratory behaviors are more likely to yield incorrect answers. This dynamic continually suppresses the model's propensity to explore diverse answers.

Algorithm 2 directly computes the factor $S_* - \mathbb{E}_{i\sim\mathbf{p}}[S_i]$, and masks those who contribute extremely significantly to the covariance expression.

For example, for negative samples where $A < 0$, Theorem 3.3 masks those tokens with large negative $S_* - \mathbb{E}_{i\sim\mathbf{p}}[S_i]$, who contributes a large negative factor to equation 20, stabling the change of entropy.

Algorithm 1 estimates this factor in a batch perspective, achieving better computational efficiency.

**Discussions about Parameter Sharing.** In Corollary C.2, the parameter updates induced by different tokens are linearly superimposed. While it is worth noting that in practical LLM training, parameters are shared across tokens and the global update dynamics involve complex coupling effects, establishing a rigorous theoretical model for such high-dimensional parameter interference remains an open challenge in the field of machine learning theory. Following the context of previous research (Yu et al., 2025; He et al., 2025; Su et al., 2025; Ren & Sutherland, 2025; Cui et al., 2025; Liao et al., 2025; Wang et al., 2025), our framework focuses on the microscopic atomic unit of this process, the single-token update, which serves as the fundamental building block of the global dynamics. In standard first-order optimization (e.g., SGD or Adam), the total gradient is the accumulation of individual token gradients. Under the regime of small learning rates characteristic of fine-tuning, the superposition of these single-token effects constitutes the dominant factor driving the entropy dynamics, while higher-order inter-token coupling effects are implicitly handled by the optimizer. Our empirical observations in Figure 1 and Figure 5 strongly corroborate this: the entropy shifts trend predicted by our decoupled single-token analysis ($S_*$) accurately match the actual batch-wise training dynamics, and the overall trend of entropy change in standard RFT are correctly predicted, suggesting that the first-order approximation effectively captures the primary mechanism of entropy evolution despite the underlying parameter sharing.

## D. Extension to off-policy scenarios

The derivation of Theorem 3.3 is based on the general GRPO formulation and is not restricted to the on-policy setting. In this section, we extend Corollaries 3.4, 3.5, C.1 and C.2 to the off-policy scenario. When off-policy sampling is used, similar expressions can be obtained by utilizing the importance ratio $r = \pi_\theta/\pi_{\theta_{\mathrm{sample}}}$.

**Corollary 3.4.1.** *To a first-order approximation, the expected entropy change factor $r(S_* - \mathbb{E}_{i\sim\mathbf{p}}[S_i])$ of a token within GRPO optimization is zero, i.e.,*

$$\mathbb{E}_{k\sim\mathbf{p}'}\left[r(S_* - \mathbb{E}_{i\sim\mathbf{p}}[S_i])\right] = 0,$$

where $\mathbf{p}'$ and $\mathbf{p}$ denote the sampling policy's and the current policy model's output distributions at token $t$, respectively.

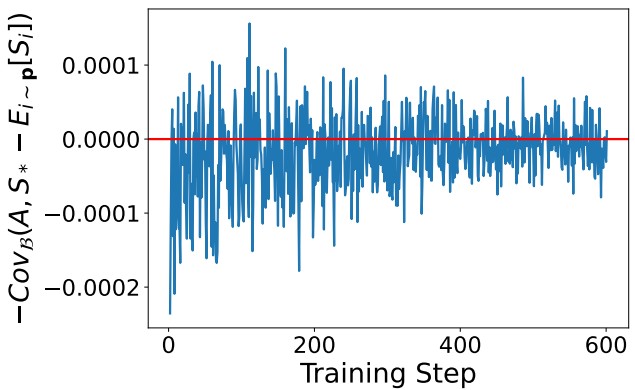

*Figure 5.* The value of $-\text{Cov}_{\mathcal{B}}(A, S_* - \mathbb{E}_{i\sim\mathbf{p}}[S_i])$.

*Proof.* We derive the results as follows:

$$\mathbb{E}_{k\sim\mathbf{p}'}[r(S_* - \mathbb{E}_{i\sim\mathbf{p}}[S_i])] = \mathbb{E}_{k\sim\mathbf{p}'}\left\{ r\left[ p_k(H + \log p_k) - \sum_{i=1}^{V} p_i^2(H + \log p_i) \right] \right\}$$

$$= \sum_{k=1}^{V} \frac{p_k}{p_k'} p_k' p_k(H + \log p_k) - \sum_{i=1}^{V} p_i^2(H + \log p_i) \sum_{k=1}^{V} \frac{p_k}{p_k'} p_k'$$

$$= (1 - \sum_{k=1}^{V} p_k) \sum_{i=1}^{V} p_i^2(H + \log p_i)$$

$$= 0.$$

$\square$

**Corollary 3.5.1.** *For on-policy GRPO training with a batch, the expected value of entropy change factor $r(S_*^t - \mathbb{E}_{i\sim\mathbf{p}_t}[S_i^t])$ over the batch of tokens $\mathcal{T}_{\mathcal{B}}$ is zero:*

$$\mathbb{E}_{t\in\mathcal{T}_{\mathcal{B}}}[r(S_*^t - \mathbb{E}_{i\sim\mathbf{p}_t}[S_i^t])] = 0. \tag{24}$$

*Proof.* For each token in Batch $\mathcal{T}_{\mathcal{B}}$, define $\mathbf{p}_t$ as the distribution of the current policy, and $\mathbf{p}_t'$ as the distribution of the sampling policy. Considering one time step $t$, the selected token $K_t$ follows the distribution $\mathbf{p}_t'$, i.e., $K_t \sim \mathbf{p}_t'$. The importance ratio of token $K_t$ can be expressed as $r = \frac{\mathbf{p}_t(K_t)}{\mathbf{p}_t'(K_t)}$. The conditional expectation of each term under the sampling distribution $\mathbf{p}_t'$ is then given by:

$$\mathbb{E}_{K_t\sim\mathbf{p}_t'}\left[ r \cdot (S_*^t - \mathbb{E}_{i\sim\mathbf{p}_t}[S_i^t]) \mid \mathbf{p}_t, \mathbf{p}_t' \right].$$

We expand this expression according to the definition of expectation:

$$\mathbb{E}_{K_t\sim\mathbf{p}_t'}\left[ r \cdot (S_*^t - \mathbb{E}_{i\sim\mathbf{p}_t}[S_i^t]) \mid \mathbf{p}_t, \mathbf{p}_t' \right] = \sum_{k\in V} \mathbf{p}_t'(k) \cdot \frac{\mathbf{p}_t(k)}{\mathbf{p}_t'(k)} \cdot (S_k^t - \mathbb{E}_{i\sim\mathbf{p}_t}[S_i^t])$$

$$= \sum_{k\in V} \mathbf{p}_t(k) \cdot (S_k^t - \mathbb{E}_{i\sim\mathbf{p}_t}[S_i^t])$$

$$= \mathbb{E}_{k\sim\mathbf{p}_t}\left[ S_k^t - \mathbb{E}_{i\sim\mathbf{p}_t}[S_i^t] \right].$$

According to Corollary 3.4, under the current policy distribution $\mathbf{p}_t$, the expectation of the difference between the discriminator score $S_*$ and its expectation is 0:

$$\mathbb{E}_{k\sim\mathbf{p}_t}[S_k^t] - \mathbb{E}_{k\sim\mathbf{p}_t}[\mathbb{E}_{i\sim\mathbf{p}_t}[S_i^t]] = \mathbb{E}_{i\sim\mathbf{p}_t}[S_i^t] - \mathbb{E}_{i\sim\mathbf{p}_t}[S_i^t] = 0.$$

Therefore, for any token $K_t$ in the Batch, the expected conditioned value of the entropy change factor is 0:

$$\mathbb{E}_{K_t\sim\mathbf{p}_t'}\left[ r \cdot (S_*^t - \mathbb{E}_{i\sim\mathbf{p}_t}[S_i^t]) \mid \mathbf{p}_t, \mathbf{p}_t' \right] = 0 \tag{25}$$

Finally, taking the mean over Batch $\mathcal{T}_\mathcal{B}$ utilizing the linearity of expectation and tower property:

$$\mathbb{E}_{t\in\mathcal{T}_\mathcal{B}}[r(S_*^t - \mathbb{E}_{i\sim\mathbf{p}_t}[S_i^t])] = \mathbb{E}_{K_t\sim\mathbf{p}_t'}\left[\frac{1}{|\mathcal{T}_\mathcal{B}|}\sum_{t\in\mathcal{T}_\mathcal{B}} r(S_t^{K_t} - \mathbb{E}_{i\sim\mathbf{p}_t}[S_i^t]\,\big|\,\{\mathbf{p}_t\}_{t\in\mathcal{T}_\mathcal{B}})\right] = \frac{1}{|\mathcal{T}_\mathcal{B}|}\sum_{t\in\mathcal{T}_\mathcal{B}} 0 = 0.$$

$\square$

To extend the off-policy version of Corollaries C.1 and C.2, we leverage similar methods in proving Corollaries 1.1 2.1, i.e., replacing the results in Corollaries 3.4 and 3.5 with their off-policy counterparts. The following corollaries show the off-policy extensions of Corollaries C.1 and C.2.

**Corollary C.1.1.** *During GRPO, the first-order expectation of token entropy change is given by:*

$$\mathbb{E}_{k\sim\mathbf{p}'}[\Delta H] = -\eta\,\mathrm{Cov}_{k\sim\mathbf{p}'}(A, r(S_* - E_{i\sim\mathbf{p}}[S_i])), \tag{26}$$

where $\mathbf{p}'$ and $\mathbf{p}$ denote the sampling policy's and the current policy model's output distributions at token $t$, respectively.

**Corollary C.2.1.** *Within an GRPO training batch, the first-order expectation of entropy change is given by:*

$$\Delta H_{\mathcal{T}_\mathcal{B}} = -\eta\,\mathrm{Cov}_\mathcal{B}(A, r(S_* - E_{i\sim\mathbf{p}}[S_i])). \tag{27}$$

## E. Supplemental results of the experiment

### E.1. Detailed training Curves

The training dynamic of average@K accuracy and Entropy in Table 1 are provided in Figure 6.

### E.2. Experiments with PPO

We provide a simple demonstration with PPO on Qwen2.5-7B-Instruct model. We directly apply the GAE Advantage from PPO as the criterion for determining the token optimization direction in our algorithms, i.e.

$$\delta_t = r_t + \gamma V_{t+1} - V_t\,,$$

$$A_t = \delta_t + (\gamma\lambda)A_{t+1}\,,$$

where $V$ denotes the state value assigned by critic model, $\gamma$ denotes the discount factor and $\lambda$ represents smoothing parameter. As a simple and direct application to PPO, our methods achieve significant improvements, as shown in table 3.

*Table 3.* Experiment results of $\mathrm{Clip}_\mathcal{B}/\mathrm{Clip}_\mathcal{V}$ with PPO training algorithm.

| Method | AIME24 | AIME25 | DAPO500 |
|---|---|---|---|
| Vanilla PPO | 16.15 | 13.75 | 40.98 |
| $\mathrm{Clip}_\mathcal{B}$ | 16.56 | **15.31** | **46.12** |
| $\mathrm{Clip}_\mathcal{V}$ | **17.60** | 15.21 | 44.50 |

We believe this result convincingly demonstrates the potential of our work to be applied across various policy gradient methods and highlights that developing entropy control methods tailored for different RFT algorithms based on the entropy dynamics is a promising direction for future work.

### E.3. Experiments with More Models

We conduct additional experiments on Qwen3-4B-Base (hereafter mentioned as Qwen3), DeepSeek R1-Distill-llama-8B-Instruct (hereafter mentioned as Distilled-Llama), and InternLM3-8B-Instruct (hereafter mentioned as InternLM). The average@K performance of the models is listed in Table 4, and the training dynamics are provided in Figure 7.

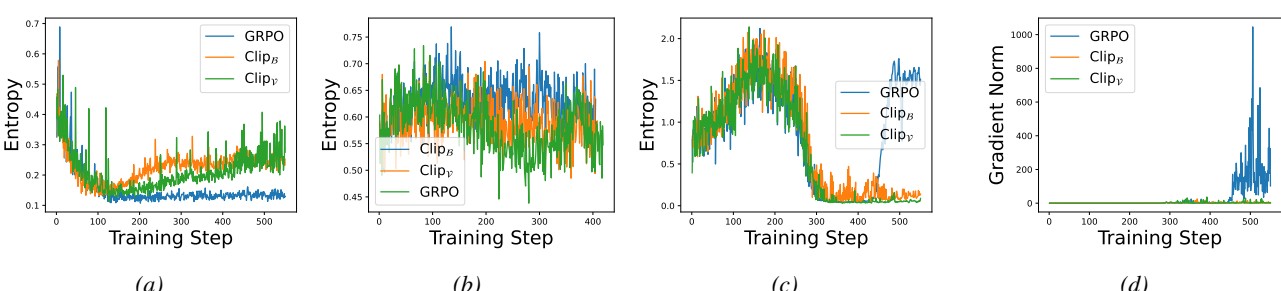

*Figure 6.* Full curves of performance and entropy for different models.

*Figure 7.* Dynamics of entropy during RFT of Qwen3(a), Distilled-llama(b), Internlm(c) and gradient norm of Internlm(d).

*Table 4.* Avg@K accuracy of models trained from more base models.

| Method | AIME24 | AIME25 | DAPO500 |
|---|---|---|---|
| Qwen3-4B-Base | 9.06 | 7.71 | 27.82 |
| GRPO | 20.73 | 18.96 | 50.38 |
| GRPO+$\text{Clip}_{\mathcal{B}}$ | 20.83 | 19.48 | 49.70 |
| GRPO+$\text{Clip}_{\mathcal{V}}$ | **21.56** | **20.42** | **50.95** |
| R1-Distill-llama-8B-Instruct | | | |
| GRPO | 31.87 | 23.85 | 60.12 |
| GRPO+$\text{Clip}_{\mathcal{B}}$ | 32.19 | 24.37 | **60.42** |
| GRPO+$\text{Clip}_{\mathcal{V}}$ | **32.40** | **24.69** | 59.88 |
| InternLM3-8B-Instruct | 4.38 | 3.43 | 19.30 |
| GRPO | 9.17 | 6.35 | 29.50 |
| GRPO+$\text{Clip}_{\mathcal{B}}$ | **9.27** | **7.29** | **30.88** |
| GRPO+$\text{Clip}_{\mathcal{V}}$ | 8.85 | 6.77 | 30.23 |

As listed in Table 4, our methods outperform baselines in most scenarios, demonstrating that their effectiveness in encouraging exploration and improving model performance can generalize across different models.

The training dynamics exhibited in Figure 7 vary across different models. For Qwen3, the training dynamics are similar to those of the Qwen2.5 series models. Our methods effectively alleviate the entropy collapse phenomenon. For Distilled-Llama, although the model's training dynamics differ significantly from those of Qwen, our method still demonstrates strong entropy-stabilizing properties and achieves competitive model performance. In the training of InternLM, our method demonstrates useful benefits in stabilizing training. Despite employing additional data filtering and hyperparameter tuning, InternLM consistently suffers from training collapse when using Vanilla GRPO. In contrast, our method enables stable and sustained training. The corresponding training dynamics are shown in Figure 7d: Vanilla GRPO exhibits significant gradient fluctuations in the later stages of training, whereas our method remains relatively stable. This suggests that our filtering of outlier tokens also contributes to training stability.

