# OpenReview forum: "On the Entropy Dynamics in Reinforcement Fine-Tuning of Large Language Models"
_ICML.cc/2026/Conference — ICML 2026 regular_

### Official Review · Reviewer_Wgzh · 2026-03-11

**Soundness:** 2
**Presentation:** 3
**Significance:** 3
**Originality:** 3
**Overall Recommendation:** 5
**Confidence:** 3

**Summary:**

The paper set down an analysis of the change in the entropy dynamics during the RFT, first quantifying the entropy change under a single logit update, then focusing on the GRPO update to derive a "entropy clipping" method, namely a way to clip  so that the induced entropy does not drift excessively. The paper then proposes a series of experiments to investigate the effect of this sort of entropy-aware regularization.

**Compliance With Llm Reviewing Policy:**

Affirmed.

**Final Justification:**

the rebuttal addressed all my concerns, I believe by adding the requested results/considerations the paper will be ready for publication.

**Key Questions For Authors:**

- Can the authors address my raised concerns about the significance of the work? How can results for GRPO in Theorem 3.3 be used to extract other considerations about other methods?
- Can the authors articulate on what do they mean by discriminator in this context and try to rephrase the intuition behind it?

**Limitations:**

I was not able to find any argument of the limitations of the proposed method unfortunately.

**Strengths And Weaknesses:**

**Soundness**:
The theoretical analysis seems sound and intuitively justifies the proposed regularization starting from GRPO.  Unfortunately, in my understanding the theoretical considerations from Th 3.3 on are valid for GRPO-based algorithms, and I believed that considerations in section 4.2 about other regularization techniques might be explained better, namely: how theorem 3.3. can be used to derive considerations for non-GRPO methods. Finally, the paper provides empirical evidence about the role of the proposed regularization by itself or against GRPO, but it does not compare it against other regularization methods.
In other words, while the proposed regularization is justified, I was not able to extract formal support for the actual advantages of it against other existing approaches.

**Presentation**:
the paper is generally well written, I have no comments on this side.

**Significance**:
I believe the proposed method to be significant in general, as the investigation of specific regularization techniques motivated by and leveraging RL over LLM specific structures is of fundamental interest.

**Originality**:
I believe this work to reach a good level of originality.

---

> ### Author Rebuttal · Authors · 2026-03-31
>
> Thank you for the feedback. We respond point by point below.
>
> ---
>
> > Q1: ...from Th 3.3 on are valid for GRPO-based algorithms...considerations in Section 4.2 about other regularization techniques might be explained better...
>
> Thank you for your question. The derivations in Sec 3.2, including Thm 3.3 and Cor. 3.4 and 3.5, are NOT specific to GRPO. They analyze the entropy dynamics induced by the core RFT surrogate loss (Eq. 8), which is shared by mainstream policy gradient methods such as GRPO, PPO, DAPO, and GSPO. Therefore, the analysis naturally extends to these algorithms. We use GRPO as the main example because it underlies many related methods and makes the presentation clearer.
>
> In Sec 4.2, we discuss several algorithms developed on GRPO or PPO. These methods incorporate the aforementioned core surrogate loss, while introducing additional modifications such as weighting and clipping. Accordingly, we analyze how these modifications affect the entropy dynamics revealed by Thm 3.3 to better understand the mechanisms behind their improvements.
>
> In App E.2, we further present results on applying our proposed entropy control method to PPO, providing additional evidence for the generality of our theoretical analysis.
>
> > Q2: ...it does not compare it against other regularization methods...
>
> Following your suggestion, we add comparisons with entropy regularization baselines [1–3] and additional datasets. These results (avg@K) further show that our methods ClipB and ClipV are competitive with existing methods. We will update the results on Qwen2.5-14B after the experiments are completed.
>
> |Model-7B|AIME 24|AIME 25|DAPO 500|AMC23|Minerva|Olympaid|Avg|
> |-|-|-|-|-|-|-|-|
> |Vanilla GRPO|16.88|15.42|48.03|61.64|54.83|53.41|41.70|
> |Clip-Cov[1]|16.67|15.00|47.98|*62.11*|55.11|*54.81*|41.95|
> |KL-Cov[1]|17.40|**16.77**|46.60|60.39|56.53|53.99|41.95|
> |Rewarding Unlikely[2]|15.94|15.83|48.22|56.25|*57.67*|51.64|40.93|
> |Forking tokens[3]|16.36|14.90|47.13|59.30|53.98|53.45|40.85|
> |ClipB|**19.69**|*16.36*|**49.68**|**62.66**|**58.52**|**55.64**|**43.76**|
> |ClipV|*18.12*|15.94|*49.65*|61.95|57.39|54.76|*42.97*|
>
> > Q3 ...what do they mean by discriminator...and...the intuition behind it?
>
> **The Meaning of the Discriminator**
>
> The discriminator $S_*$ is the key quantity that determines the **direction of entropy change** induced by parameter update associated with a token.
>
> For a single-logit update (Thm 3.2), the direction of entropy change is determined by the sign of $\varepsilon S_*$. The sign of $\varepsilon$ is fully determined by whether the token is rewarded or penalized. With $\varepsilon$ fixed, the sign of $S_*$ tells us the direction of the entropy change. For this reason, we refer to $S_*$ as a discriminant quantity.
>
> In practical RFT algorithms (Thm 3.3), the entropy change is determined by the sign of $\alpha (S\_* - \mathbb{E}\_{i \sim p}[S_i])$. Since the sign of $\alpha$ depends on whether the sample is positive or not, the direction of entropy change is governed by the difference between $S_*$ and its expectation $\mathbb{E}\_{i \sim p}[S_i]$. Clearly, $S_*$ is the key quantity for determining the direction of entropy change.
>
> **The Intuition Behind the Discriminator**
>
> Prior work[2] observed that tokens with different probabilities have diverse impacts on the change of model's entropy. Intuitively, high-probability positive samples tend to decrease entropy, while low-probability positive samples tend to increase it; for negative samples, the trend is reversed. Related observations also appear in contemporaneous work [4][5].
>
> Motivated by empirical observations, we aim to provide a precise and clean discriminator that identifies the probability boundary separating entropy-increasing and entropy-decreasing updates for positive and negative tokens, which can support more principled algorithm design and theoretical analysis.
>
> $S_*$ fits this role well, as it is computed from two common quantities, entropy and probability, causing almost no additional computational overhead. Moreover, the experiments in Figure 1 show that it is sufficiently accurate in discriminating the direction of entropy change.
>
> ---
>
> Thank you again for your suggestions! We have added the above discussions and results to manuscript. We hope our response can address your problems and lean you more torwards the acceptance of this work.
>
> References:
>
> [1]Cui et al. The Entropy Mechanism of Reinforcement Learning for Reasoning Language Models
>
> [2]He et al. Rewarding the Unlikely
>
> [3]Wang et al. Beyond the 80/20 Rule
>
> [4]Hao et al. Rethinking Entropy Interventions in RLVR: An Entropy Change Perspective
>
> [5]Xi et al. BAPO

---

> > ### Author Rebuttal · Reviewer_Wgzh · 2026-04-03
> >
> > Fully resolved, raised to 5. One **important** thing: as of today, I believe this paper nicely relates to the ICLR 2026 paper "Entropy-preserving reinforcement learning". I think it would be *great* to add an extensive discussion of the differences.

---

> > > ### Author Response · Authors · 2026-04-07
> > >
> > > Dear reviewers,
> > >
> > > We sincerely appreciate your positive feedback and your updated assessment. We are very encouraged that our rebuttal has adequately addressed your concerns.
> > >
> > > Following your latest suggestion, we have added a discussion of *Entropy-preserving Reinforcement Learning* to the revised related-work section and clarified the distinctions between their perspective and ours.
> > >
> > > We have now completed training the 14B models, and we are providing the results here. The results further confirmed the advantage of our methods.
> > >
> > > | Model-14B      | AIME24    | AIME25    | DAPO500   | AMC23     | Minerva   | Avg       |
> > > | -------------- | --------- | --------- | --------- | --------- | --------- | --------- |
> > > | Vanilla GRPO   | 22.50     | 17.60     | 52.95     | 66.33     | 68.47     | 45.57     |
> > > | Clip-Cov       | 22.97     | *20.62*   | 59.85     | **70.31** | 69.03     | 48.56     |
> > > | KL-Cov         | 22.19     | 19.95     | 59.05     | 69.53     | *70.45*   | 48.23     |
> > > | Forking Tokens | 21.15     | 20.42     | 58.15     | 67.58     | 69.60     | 47.38     |
> > > | ClipB          | *23.33*   | *20.62*   | *60.35*   | *70.16*   | 69.60     | *48.81*   |
> > > | ClipV          | **23.44** | **21.35** | **61.92** | 69.45     | **71.02** | **49.44** |
> > >
> > > Thank you again for your constructive comments, which have helped us significantly improve both the positioning and the empirical completeness of the paper. We hope this further strengthens your confidence in accepting our work.
> > >
> > > ---
> > >
> > > Best regards,
> > >  Authors

---

### Official Review · Reviewer_grja · 2026-03-12

**Soundness:** 2
**Presentation:** 3
**Significance:** 3
**Originality:** 2
**Overall Recommendation:** 4
**Confidence:** 3

**Summary:**

This paper focuses on the entropy aspects of RL fine-tuning. In specifics, authors argue that entropy is utilized to achieve a balance between exploration and exploitation operations. However, previous works ignore the learning dynamics of entropy-guided RL. Specifically, authors derive the expression for the token-level entropy change during policy optimization and utilize these derived theoretical results to add some controls into GROP for stabilizing the training process.

**Compliance With Llm Reviewing Policy:**

Affirmed.

**Final Justification:**

authors have fully resolved my concerns. they have added more benchmarks to validate their proposed framework. so i raise the score to 4.

**Key Questions For Authors:**

1. Since there already exists one paper [1] that also focuses on entropy control for LLM-RL, what is the specific difference between yours and theirs? If possible, please make more clarifications and explanations.

2. In your experiments, the utilized datasets are AIME24, AIME25 and DAPO500. In [2], its authors conduct experiments on benchmarks MATH-Hard, MATH-500, AIME24, Minerva, Olympiad and AMC to conduct experiments. Since [2] is a highly relevant work, can you please also add results on these extra datasets to show the utility of your proposed framework in a more comprehensive setting?


[1] Shen, Han. "On entropy control in llm-rl algorithms." arXiv preprint arXiv:2509.03493 (2025).

**Limitations:**

1. Authors didn't go through all of the revelant papers in a complete and thorough manner. I suggest authors expand their related work and references.

2. The current conducted experiments are not comprehensive. I suggest authors add more results of related benchmarks to show the availabilty of proposed framework.

**Strengths And Weaknesses:**

Strengths And Weaknesses:

1. Paper is clearly written. The motivation is well formulated.

2. The entropy dynamic view is novel. It brings some new insights into this research direction.

3. The paper is full of theoretical analysis. It makes the whole paper very sound.

4. However, the related works and references seem incomplete. For instance, [1] also cares about the token-level entropy and they find that thinking tokens are very informative and will make huge influences to the reasoning results. Besides, there are also many other very important and relevant references like [2,3,4]. Especially for [4], they also mention the concept of entropy control.


[1] Qian, Chen, et al. "Demystifying Reasoning Dynamics with Mutual Information: Thinking Tokens are Information Peaks in LLM Reasoning." The Thirty-ninth Annual Conference on Neural Information Processing Systems.

[2] Cui, Ganqu, et al. "The entropy mechanism of reinforcement learning for reasoning language models." arXiv preprint arXiv:2505.22617 (2025).

[3] Cheng, Daixuan, et al. "Reasoning with exploration: An entropy perspective." arXiv preprint arXiv:2506.14758 (2025).

[4] Shen, Han. "On entropy control in llm-rl algorithms." arXiv preprint arXiv:2509.03493 (2025).

---

> ### Author Rebuttal · Authors · 2026-03-31
>
> Thank you for your feedback. We would like to answer your questions point to point to address your concerns.
>
> ---
>
> > Q1: Since there already exists one paper [1] that also focuses on entropy control for LLM-RL, what is the specific difference between yours and theirs? If possible, please make more clarifications and explanations.
>
> Our work and [1] focus on fundamentally different problems. Qian et al. [1] studies the reasoning trajectory of LLM at **inference** from an information-theoretic perspective: they track the  mutual-information (MI) between intermediate hidden representations and the gold answer, identify sparse MI peaks / thinking tokens, and use these observations to design training-free inference-time methods.
>
> In contrast, our paper studies entropy dynamics during the **training** of reinforcement fine-tuning. Our analysis is conducted directly in the logit space and answers a different question: how a token update in RFT/GRPO changes the model’s output entropy, and why entropy collapse happens during RL training.
>
> [1]Demystifying Reasoning Dynamics with Mutual Information: Thinking Tokens are Information Peaks in LLM Reasoning
>
> > Q2: ...Since Cui et al. is a highly relevant work, can you please also add results on these extra datasets to show the utility of your proposed framework in a more comprehensive setting?
>
> > L2: The current conducted experiments are not comprehensive. I suggest authors add more results of related benchmarks to show the availabilty of proposed framework.
>
> Thank you for your suggestions. We choose AIME24, AIME25 and DAPO500 because they are relatively new and challenging compared to earlier datasets such as MATH, MATH500 and GSM8k, providing less chance of data coruption by the base models.
>
> Following DAPO, all datasets we choose(including DAPO-MATH-17k for training) contain only integer answers, making the evaluation more accurate thus providing more clean, stable and convincing enviroment for our research.
>
> We now provide avg@K accuracy results on AMC23, interger set of inerva-Math and OlympaidBench, algong with four more baselines. Results on Qwen2.5-14B will be updated as soon as possible.
>
> |Model-7B|AIME 24|AIME 25|DAPO 500|AMC23|Minerva|Olympaid|Avg|
> |-|-|-|-|-|-|-|-|
> |Vanilla GRPO|16.88|15.42|48.03|61.64|54.83|53.41|41.70|
> |Clip-Cov[1]|16.67|15.00|47.98|*62.11*|55.11|*54.81*|41.95|
> |KL-Cov[1]|17.40|**16.77**|46.60|60.39|56.53|53.99|41.95|
> |Rewarding Unlikely[2]|15.94|15.83|48.22|56.25|*57.67*|51.64|40.93|
> |Forking tokens[3]|16.36|14.90|47.13|59.30|53.98|53.45|40.85|
> |ClipB|**19.69**|*16.36*|**49.68**|**62.66**|**58.52**|**55.64**|**43.76**|
> |ClipV|*18.12*|15.94|*49.65*|61.95|57.39|54.76|*42.97*|
>
> The results demosntrate that our methods achive competitive performance compared to existing practical methods, cofirming the advantage of them.
>
> References:
>
> [1] Cui et al. The Entropy Mechanism of Reinforcement Learning for Reasoning Language Models
>
> [2] He et al. Rewarding the Unlikely: Lifting GRPO Beyond Distribution Sharpening
>
> [3] Wang et al. Beyond the 80/20 Rule
>
> > L1: Authors didn't go through all of the revelant papers in a complete and thorough manner. I suggest authors expand their related work and references.
>
> Thank you for your suggestion. In the revised Related Work section, we have incorporated all the papers you recommended, along with several recent studies on entropy collapse. The newly added works include:
>
> 1. Studies on the relationship between reasoning model entropy and model performance [1] [2] [3] [4]
> 2. Studies on practical techniques for mitigating entropy collapse [5] [6] [7] [8] [9]
> 3. Studies investigate the patterns of entropy change during RFT [10] [11] [12] [13]
>
> A shortened version is given in the Response to Reviewer Q3pb, due to the space limit.
>
> References:
>
> [1] Yue et al. Does Reinforcement Learning Really Incentivize Reasoning Capacity in LLMs Beyond the Base Model?
>
> [2] Jin et al. Revisiting Entropy in Reinforcement Learning for Large Reasoning Models
>
> [3] Qian et al. Demystifying Reasoning Dynamics with Mutual Information: Thinking Tokens are Information Peaks in LLM Reasoning
>
> [4] Jin et al. Revisiting Entropy in Reinforcement Learning for Large Reasoning Models
>
> [5] Jiang et al. Rethinking Entropy Regularization in Large Reasoning Models
>
> [6] Cheng et al. Reasoning with Exploration: An Entropy Perspective
>
> [7] Shen et al. On Entropy Control in LLM-RL Algorithms
>
> [8] Chen et al. Pass@k Training for Adaptively Balancing Exploration and Exploitation of Large Reasoning Models
>
> [9] Su et al. CE-GPPO
>
> [10] Cui et al. The Entropy Mechanism of Reinforcement Learning for Reasoning Language Models
>
> [11] Hao et al. Rethinking Entropy Interventions in RLVR: An Entropy Change Perspective
>
> [12] Xi et al. BAPO
>
> [13] Petrenko et al. Entropy-Preserving Reinforcement Learning
>
>
> ---
>
> Thank you again for your suggestions! We hope our response can address your concerns and lean you more torwards the acceptance of our work.

---

> > ### Author Rebuttal · Reviewer_grja · 2026-04-03
> >
> > The authors' new submitted rebuttals have resolved my concerns. I therefore raise the score to 4.

---

> > > ### Author Response · Authors · 2026-04-07
> > >
> > > Dear reviwers,
> > >
> > > We sincerely appreciate your positive feedback and your updated assessment. We are especially grateful for your suggestion to broaden the datasets, empirical comparison and discussion of related work, which helped us substantially strengthen the paper.
> > >
> > > We have now completed the experiments on Qwen2.5-14B and provide the results below. These new results further support the effectiveness of our methods across scales and benchmarks.
> > >
> > > | Model-14B      | AIME24    | AIME25    | DAPO500   | AMC23     | Minerva   | Avg       |
> > > | -------------- | --------- | --------- | --------- | --------- | --------- | --------- |
> > > | Vanilla GRPO   | 22.50     | 17.60     | 52.95     | 66.33     | 68.47     | 45.57     |
> > > | Clip-Cov       | 22.97     | *20.62*   | 59.85     | **70.31** | 69.03     | 48.56     |
> > > | KL-Cov         | 22.19     | 19.95     | 59.05     | 69.53     | *70.45*   | 48.23     |
> > > | Forking Tokens | 21.15     | 20.42     | 58.15     | 67.58     | 69.60     | 47.38     |
> > > | ClipB          | *23.33*   | *20.62*   | *60.35*   | *70.16*   | 69.60     | *48.81*   |
> > > | ClipV          | **23.44** | **21.35** | **61.92** | 69.45     | **71.02** | **49.44** |
> > >
> > > We believe that, with your help, the quality of our work has improved significantly. We hope this further strengthens your confidence in accepting our work.
> > >
> > > ---
> > >
> > > Best regards,
> > >  Authors

---

### Official Review · Reviewer_Q3pb · 2026-03-12

**Soundness:** 3
**Presentation:** 2
**Significance:** 3
**Originality:** 3
**Overall Recommendation:** 4
**Confidence:** 2

**Summary:**

This paper investigates why LLMs fine-tuned with RL usually suffered from entropy collapse. The authors establish a theoretical framework to characterize entropy dynamics during policy optimization.

**Compliance With Llm Reviewing Policy:**

Affirmed.

**Final Justification:**

The authors fully resolved my concerns on Fig 1, lack of entropy dynamics plots, and missing related works. In general, this paper is acceptable. The reason I only provide suggestion of weak accept is due to the foundation of this work --- studying entropy dynamics and entropy-controlled policy optimization. In the setting of LLMs, entropy might not be a good metric to reflect the health of policy as it's dependent on lots of factors such as response length. However, this problem is not solvable by rebuttal, as it is the theme of this paper. Based on this reason, I can only provide a rate 4.

**Key Questions For Authors:**

I might be missing something, but why Figure 1 (a) and (b) have exactly the same curves but only different colors?

**Limitations:**

- The paragraph of Entropy Dynamics (line 70) is not very accurate. a_t is never defined.
- Figure 2, (a) and (b) lack baselines
- Shouldn't the author have an experiment to show entropy dynamics of the proposed method and baselines?
- More related works on addressing entropy collapse should be discussed.

**Strengths And Weaknesses:**

- Soundness: This paper is in general sound. The first-order analysis on the entropy dynamics are understandable and interesting. The discriminator has also been shown via experiments.
- Presentation is generally fine. Some of the writings can be improved (see limitations).
- Significance: I think theoretical entropy analysis and a principled algorithm to address the collapse problem have been lacking. So this paper address this.
- Originality is fine to me. But I might be missing some literature. But the related work section should be expanded.

---

> ### Author Rebuttal · Authors · 2026-03-31
>
> Thank you for your feedback. We would like to answer your questions point to point to address your concerns.
>
> ---
>
> > Q1: why Figure 1 (a) and (b) have exactly the same curves?
>
> Curves in Figure 1(a) and (b) are NOT the same. Please pay particular attention to the trend between step 10 and 20, as well as the differences in Y-axis ticks. We provide some raw statistics of entropy to make it clearer.
> |Experiment|Step5|Step10|Step15|Step20|
> |-|-|-|-|-|
> |Retain S*<0|0.289|0.253|0.142|0.102|
> |Retain S*>0|0.388|0.410|0.611|1.015|
> |Baseline|0.345 |0.301|0.273|0.284|
> |Mask S*<0 |0.324 |0.221|0.140|0.132|
> |Mask S*>0 |0.363 |0.450|0.730|1.038|
>
> > L1: a_t is not defined.
>
> a_t is simply the notation of the sampled token at position t, as explicitly defined in the sentence on line 70-77. We will refine the sentence to make this clearer.
>
> > L2: Figure 2(a)(b) lack baseline.
>
> These figures show how the value of $\mu$ affects the fraction of tokens being clipped by our algorithms. For the baseline without our proposed method, the clipping fraction remains zero throughout. We will add the baseline curves to the figures.
>
> > L3: entropy dynamics of the proposed method and baselines?
>
> The entropy dynamics are already provided in the curves in Figure 6 of Appendix E.
>
> > L4: More related works on addressing entropy collapse should be discussed.
>
> Thank you for your suggestion. We have added some latest works on entropy collapse and revised our related work section. A shortened version is given below:
>
> ```
> Reinforcement fine-tuning (RFT) has been widely adopted for tuning LLMs, including representative methods such as GRPO[1], DAPO[2], and GSPO[3].
>
> Early studies [4][5] observed that the exploratory capacity of LLMs gradually diminishes during RFT, often manifested as entropy collapse [2]. To better understand this phenomenon, several works have examined the link between token entropy and reasoning performance. For example, He et al. [6] shows that the top 20% high-entropy tokens often act as forking tokens and thus carry greater training value, while Qian et al. [7] demonstrates from an information-theoretic perspective that a small number of thinking tokens can substantially influence reasoning performance. Jin et al. [8] further analyzes why preserving entropy benefits RFT.
>
> Another line of work aims to mitigate entropy collapse through practical techniques. Some methods introduce explicit entropy regularization [9][10][11][12] or directly optimize Pass@K as training objective [13]. Others target the underlying causes of entropy collapse based on empirical or theoretical insights, such as flexible clipping schemes [2][14] and sample weighting by token probabilities [15][16]. Liu et al. [17] systematically evaluates many of these methods across different models and provides practical guidance.
>
> Closest to our work are studies that analyze how different tokens contribute to the entropy dynamics. Cui et al. [18] links entropy changes to sampling distributions and models the performance–entropy relationship, but its method relies on the advantages of unsampled tokens, which are difficult to estimate in RFT algorithms. Hao et al. [19] and Xi et al. [20] show that tokens from positive and negative samples can be divided into four regions according to their probability ranges, which is broadly consistent with our analysis. Petrenko et al. [21] empirically studies entropy-change patterns across different RFT algorithms and provides several useful insights.
>
> References:
>
> [1] Shao et al. DeepSeekMath
> [2] Yu et al. DAPO
> [3] Zheng et al. GSPO
> [4] Yue et al. Does Reinforcement Learning Really Incentivize Reasoning Capacity in LLMs Beyond the Base Model?
> [5] Wu et al. The Invisible Leash: Why RLVR May or May Not Escape Its Origin
> [6] Wang et al. Beyond the 80/20 Rule
> [7] Qian et al. Demystifying Reasoning Dynamics with Mutual Information: Thinking Tokens are Information Peaks in LLM Reasoning
> [8] Jin et al. Revisiting Entropy in Reinforcement Learning for Large Reasoning Models
> [9] Hu et al. Open-Reasoner-Zero
> [10] Jiang et al. Rethinking Entropy Regularization in Large Reasoning Models
> [11] Cheng et al. Reasoning with Exploration: An Entropy Perspective
> [12] Shen et al. On Entropy Control in LLM-RL Algorithms
> [13] Chen et al. Pass@k Training for Adaptively Balancing Exploration and Exploitation of Large Reasoning Models
> [14] Su et al. CE-GPPO
> [15] He et al. Rewarding the Unlikely
> [16] Yang et al. Do Not Let Low-Probability Tokens Over-Dominate in RL for LLMs
> [17] Liu et al. Part I: Tricks or Traps? A Deep Dive into RL for LLM Reasoning
> [18] Cui et al. The Entropy Mechanism of Reinforcement Learning for Reasoning Language Models
> [19] Hao et al. Rethinking Entropy Interventions in RLVR: An Entropy Change Perspective
> [20] Xi et al. BAPO
> [21] Petrenko et al. Entropy-Preserving Reinforcement Learning
> ```
> ---
>
> Thank you again for your suggestions! We hope our response can address your concerns and lean you more towards the acceptance of our work.

---

> > ### Author Rebuttal · Reviewer_Q3pb · 2026-04-03
> >
> > I would like to keep my positive score.

---

> > > ### Author Response · Authors · 2026-04-07
> > >
> > > Dear reviwers,
> > >
> > > We sincerely appreciate your positive feedback and are very glad that our response has adequately addressed your concerns.
> > >
> > > During the rebuttal period, we further strengthened the paper by adding comparisons with additional entropy-control baselines, expanding the evaluation to more datasets, and completing the 14B experiments. These additions were helpful in addressing the remaining concerns on empirical completeness and led other reviewers who had raised related issues to view the paper more positively.
> > >
> > > We belive that the quality of our work has been significantly improved with your constructive suggestions. We hope these newly added results and discussions could further strengthen your confidence in the acceptance of the paper.
> > >
> > > ---
> > > Best regards,
> > >  Authors

---

### Official Review · Reviewer_wBSy · 2026-03-14

**Soundness:** 2
**Presentation:** 2
**Significance:** 2
**Originality:** 2
**Overall Recommendation:** 2
**Confidence:** 4

**Summary:**

the paper analyzes the entropy evolution dynamics during rl finetuning of llm. they show that the entropy decreases over time and study the impact of selectively masking out certain tokens during training, leading to different entropy behavior. they also show that by doing batch-normalized or vocab-normalized entropy update, this leads to downstream gains.

**Compliance With Llm Reviewing Policy:**

Affirmed.

**Key Questions For Authors:**

the flagship result is table 1 where the authors show performance improvement due to the normalized updates, but a key limitation of the technical solidity is that it does not compare with any related baselines achieving the same goal. starting from baseline grpo, there can be many other ways to preserve entropy during training, such as
- adding an entropy regularization
- adding a kl regularization
- annealing down the learning rate schedule
these are all valid and much lightweight ways of improving the entropy dynamics. if the proposed method stands out in simplicity of the tuning process and performance, it will send a stronger message.

does the performance gain come from higher entropy? it would be nice to have some numerical presentation/analysis beyond the flagship eval result, because that's the main message of the work. the fact that pass@k improvement is generally more significant than avg@k indicate this but good to confirm.

when reporting the results on pass@k and avg@k it would be nice to add error bars on the baseline results - p@k are notably higher variance than avg@k and this makes the comparison more significant.

**Limitations:**

certain limitations are discussed.

**Strengths And Weaknesses:**

strengths: it is interesting and important to understand entropy dynamics in rl finetuning better for llm, and this work is an interesting step for analysis and experiments.

weaknesses: beyond the mathematical analysis, the ensuing results are not super technically solid or interesting. the author shows that batch or vocab normalized entropy update yields better performance, but does it mean that they achieve these through preserving better entropy? if so, the paper should compare with other baselines such as adding kl divergence regularization, adjusting entropy bonus over time, to show why this particular way of preserving entropy is warranted.

---

> ### Author Rebuttal · Authors · 2026-03-31
>
> Thank you for your comments. We answer your questions point to point below.
>
> ---
>
> > Q1: ...it does not compare with related baselines...
>
> As noted in prior work[1][2][3], entropy regularization and hyperparameter tuning can preserve entropy but may introduce gradient conflicts or harm response quality. Following your suggestion, we compare our methods with widely-used entropy control baselines[1][4][5]:
>
> |Model-7B|AIME 24|AIME 25|DAPO 500|AMC23|Minerva|Olympaid|Avg|
> |-|-|-|-|-|-|-|-|
> |Vanilla GRPO|16.88|15.42|48.03|61.64|54.83|53.41|41.70|
> |Clip-Cov[1]|16.67|15.00|47.98|*62.11*|55.11|*54.81*|41.95|
> |KL-Cov[1]|17.40|**16.77**|46.60|60.39|56.53|53.99|41.95|
> |Rewarding Unlikely[4]|15.94|15.83|48.22|56.25|*57.67*|51.64|40.93|
> |Forking tokens[5]|16.36|14.90|47.13|59.30|53.98|53.45|40.85|
> |ClipB|**19.69**|*16.36*|**49.68**|**62.66**|**58.52**|**55.64**|**43.76**|
> |ClipV|*18.12*|15.94|*49.65*|61.95|57.39|54.76|*42.97*|
>
> The results demosntrate that our methods achive competitive performance compared to existing practical methods, cofirming the advantage of them.
>
> It is also worth noting that the main goal and contribution of our research is NOT proposing RFT algorithms achieving SOTA performance, but to give a theoretical framework for understanding how the entropy evolves during RFT (Sec 3), offering valuable insights to understand previous entropy conltrol methods (Sec 4.2) and inspiring furthur progress. Our algorithms, as straightforward applications of our theorem, serve as demonstrations of how the framework can be leveraged. And we also welcome further research on algorithm design from this perspective.
>
> References:
>
> [1] Cui et al. The Entropy Mechanism of Reinforcement Learning for Reasoning Language Models
>
> [2] Jiang et al. Rethinking Entropy Regularization in Large Reasoning Models
>
> [3] Yu et al. DAPO: An Open-Source LLM Reinforcement Learning System at Scale
>
> [4] He et al. Rewarding the Unlikely: Lifting GRPO Beyond Distribution Sharpening
>
> [5] Wang et al. Beyond the 80/20 Rule: High-Entropy Minority Tokens Drive Effective Reinforcement Learning for LLM Reasoning
>
> > Q2: does the performance gain come from higher entropy? it would be nice to have some numerical presentation/analysis beyond the flagship eval result...
>
> We would like to clarify that “higher entropy” alone does not tell the full story. While entropy is a useful indicator of a model’s exploratory capacity, increasing entropy itself is not our optimization objective. The key is to preserve the model’s ability to explore. Rather than naively regularizing the entropy or blindly tuning parameters, our method identifies the tokens that cause sharp drops in entropy and suppresses their influence, thereby maintainng the model’s exploratory capacity while introducing minimal interference to the optimization gradients.
>
> As for the numerical presentation/analysis, we have demonstrated that our methods shift the per-problem pass-rate distribution toward the middle regime rather than the degenerate extremes (Fig. 4). On the test set, the fraction of problems with partial success over rollouts increases from 28.3% to 31.7%, while all-fail decreases from 66.7% to 63.3%; on the validation set, the middle region increases from 62.6% to 65.2%, while all-pass decreases from 16.2% to 14.0%. This indicates that the model is able to explore a broader range of solutions and assigns non-zero success probability to a broader set of problems, instead of merely sharpening confidence on those easier ones.
>
> > Q3: when reporting the results on pass@k and avg@k it would be nice to add error bars on the baseline results...
>
> Thank you for your suggestion. We now provide the STD error of the baselines in our experiments, and the performance gain of each method for comparison:
>
> | | AIME24(avg@k) | AIME24(pass@k) | AIME25(avg@k) | AIME25(pass@k) | DAPO500(avg@k) | DAPO500(pass@k) |
> | --| --| --| --| --| --| --|
> | 7B STD error | 0.26| 1.67| 0.20| 1.92 | 0.25 | 0.75 |
> | 7B ClipB Gain | 2.81| 6.67 | 0.93| 3.33 | 1.65 | 3.4 |
> | 7B ClipV Gain | 1.24| 3.33 | 0.52| 6.67 | 1.62 | 2.2 |
> | 14B STD error | 0.16| 2.56 | 0.31| 1.92 | 0.27 | 0.12 |
> | 14B ClipB Gain | 0.83| 0.34 | 3.02| 6.67 | 7.40 | 1.6 |
> | 14B ClipV Gain | 0.94| 0.34 | 3.75| 6.67 | 8.97 | 2.6 |
>
> Based on the results, although Pass@K exhibits larger fluctuations, its performance gains are often more substantial as well. Most of the observed improvements are significantly larger than the range of metric variation. These results further confirm that our method effectively improves model performance from the perspectives of both avg@K and pass@K.
>
> ---
>
> We have updated all the revisions in our manuscript. We hope our responses would help you better understand our paper and address your concerns. We are looking forward to your feedback and further discussions.

---

### Decision · Program_Chairs · 2026-04-30

**Decision:**

Accept (regular)

**Comment:**

This paper presents a theoretical framework for analyzing entropy dynamics during reinforcement fine-tuning of LLMs, deriving a discriminant expression that characterizes the direction of entropy change under policy updates. Three of four reviewers support acceptance (scores: 5, 4, 4, 2). The theoretical contribution, connecting token-level updates to entropy evolution, is recognized as novel and insightful. During rebuttal, the authors substantially strengthened the paper by adding comparisons with multiple entropy-control baselines, expanding evaluation to six benchmarks, and providing comprehensive related work discussion. Three reviewers confirmed that the concerns have been well addressed, while the remaining reviewer did not actively engage in the rebuttal. Therefore, I would recommend the acceptance.